# Structural basis for the rescue of hyperexcitable cells by the amyotrophic lateral sclerosis drug Riluzole

David Hollingworth[1,8], Frances Thomas [1,8], Dana A. Page[2,8], Mohamed A. Fouda [2], Raquel Lopez-Rios De Castro[3,4], Altin Sula[5], Vitaliy B. Mykhaylyk [6], Geoff Kelly[7], Martin B. Ulmschneider [3] ✉, Peter C. Ruben [2] ✉ & B. A. Wallace [1] ✉

Neuronal hyperexcitability is a key element of many neurodegenerative disorders including the motor neuron disease Amyotrophic Lateral Sclerosis (ALS), where it occurs associated with elevated late sodium current ($I_{NaL}$). $I_{NaL}$ results from incomplete inactivation of voltage-gated sodium channels (VGSCs) after their opening and shapes physiological membrane excitability. However, dysfunctional increases can cause hyperexcitability-associated diseases. Here we reveal the atypical binding mechanism which explains how the neuroprotective ALS-treatment drug riluzole stabilises VGSCs in their inactivated state to cause the suppression of $I_{NaL}$ that leads to reversed cellular overexcitability. Riluzole accumulates in the membrane and enters VGSCs through openings to their membrane-accessible fenestrations. Riluzole binds within these fenestrations to stabilise the inactivated channel state, allowing for the selective allosteric inhibition of $I_{NaL}$ without the physical block of $Na^+$ conduction associated with traditional channel pore binding VGSC drugs. We further demonstrate that riluzole can reproduce these effects on a disease variant of the non-neuronal VGSC isoform Nav1.4, where pathologically increased $I_{NaL}$ is caused directly by mutation. Overall, we identify a model for VGSC inhibition that produces effects consistent with the inhibitory action of riluzole observed in models of ALS. Our findings will aid future drug design and supports research directed towards riluzole repurposing.

Riluzole (2-amino-6-(trifluoromethoxy)benzothiazole) is a well-tolerated front-line therapy for Amyotrophic Lateral Sclerosis (ALS)[1], a fatal motor neuron degenerative disease where death typically occurs 3-5 years after symptom onset[2,3]. Riluzole is not a cure for ALS, but it significantly increases life expectancy[4], with compelling evidence suggesting that it delays the effects of neuronal hyperexcitability-induced glutamate excitotoxicity[5,6] which initiates a cascade of events leading to motor neuron atrophy, muscle decay and eventual fatality[7].

[1]School of Natural Sciences, Birkbeck University of London, London, United Kingdom. [2]Department of Biomedical Physiology and Kinesiology, Simon Fraser University, Burnaby, BC, Canada. [3]Department of Chemistry, King's College London, London, United Kingdom. [4]Biological Physics and Soft Matter Group, Department of Physics, King's College London, London, United Kingdom. [5]Syngenta Crop Protection, Jealott's Hill International Research Centre, Bracknell, Berkshire, United Kingdom. [6]Diamond Light Source, Harwell Science and Innovation Campus, Chilton, Didcot, UK. [7]The Medical Research Council Biomedical NMR Centre, The Francis Crick Institute, London, UK. [8]These authors contributed equally: David Hollingworth, Frances Thomas, Dana A. Page. ✉e-mail: martin.ulmschneider@kcl.ac.uk; pruben@sfu.ca; b.wallace@bbk.ac.uk

Riluzole has been shown to interact with many potential protein targets, but most of these interactions produce effects at drug concentrations that greatly exceed what is therapeutically achieved[8]. However, at its clinically relevant concentration of $\sim 1$–$2\,\mu M$, riluzole interacts with voltage-gated sodium channels (VGSCs), inhibiting cellular excitability by specifically stabilising the non-conducting inactivated channel state[8]. In whole-cell recordings, this is reflected by riluzole causing hyperpolarisation of steady-state inactivation (SSI) and prolonging channel recovery from inactivation (RFI), with apparent dissociation constants for the interaction between riluzole with VGSCs inferred from these effects as low as $0.2$–$0.3\mu M$[9,10]. Hyperpolarisation of SSI reduces a non-inactivating late sodium current ($I_{NaL}$), which follows the fast-depolarising transient sodium current ($I_{NaT}$) of an action potential (AP). $I_{NaL}$ is abnormally increased in many hyperexcitability-associated diseases[11], and elevated $I_{NaL}$ has been described in both sporadic and familial ALS[12–14]. In models of ALS, riluzole administration reduces $I_{NaL}$ and decreases motor neuron excitability[15–17], with $<1\,\mu M$ shown to restore both to the levels found in motor neuron controls[15,16]. Unlike traditional VGSC drugs, riluzole targets $I_{NaL}$ without the concurrent inhibition of $I_{NaT}$[18,19], avoiding the adverse side effects that can be produced by these pore-binding drugs resulting from the off-target block of VGSC isoforms in healthy tissues[20,21]. Because the channel pore provides the $Na^+$ conduction pathway for both $I_{NaT}$ and $I_{NaL}$[22], riluzole must selectively inhibit $I_{NaL}$ through an atypical allosteric mechanism that involves binding outside the channel pore.

VGSCs can be classified as populating three transitional states, i.e., resting (closed), activated (open) and inactivated (closed). In a hyperpolarised membrane VGSCs exist in the resting state which, on sufficient depolarisation, causes them to activate to form the upstroke of an AP, before entering a non-conductive inactivated state from which they remain unable to pass current during periods of recovery. Inactivation occurs over both fast (millisecond) and slow (up to seconds) timescales and controls channel availability, and consequently influences cellular excitability[23,24]. Incomplete inactivation occurs when a fraction of VGSCs remain open, (or close and quickly re-open), after an AP and impacts cellular function, as the sustained $I_{NaL}$ that this produces reduces the current required to elicit an action potential, facilitating excitability. $I_{NaL}$ has a fundamental physiological role in regulating the excitability of cells and tissues and is modulated according to function. For example, in motor neurons, changes in $I_{NaL}$ are involved in enabling the amplified signals required for movement and postural control[25], while in cardiac myocytes changes in $I_{NaL}$ play a role in controlling cardiac rhythm[26]. Consequently, $I_{NaL}$ is highly regulated, either through homeostatic processes[27] or by direct interaction with VGSC co-factors (e.g., β-subunits[28] and calmodulin[29]). Failure of $I_{NaL}$ regulation that leads to abnormal elevation of this current causes cellular hyperexcitability which, due to the wide distribution of VGSCs in excitable tissue throughout the body, is associated with the development of a variety of diseases, including epilepsies, arrhythmias, myotonias and neuropathic pains[11], as well as neurodegenerative diseases such as ALS and Alzheimer's Disease[30], making selective targeting of $I_{NaL}$ a goal for drug development. Riluzole is a drug that selectively targets $I_{NaL}$ and, although it is presently only used in the treatment of ALS, it has also been investigated for potential repurposing in many of the aforementioned diseases[31–34]. However, a structural understanding of how riluzole interacts with VGSCs to produce this effect has been lacking.

In this study, we show that riluzole binds VGSCs at a site wholly contained within their intramembrane fenestrations to produce modulation of channel function without obstructing the Na+ conduction pathway vital for normal channel function and reveal the membrane-mediated mechanism that explains how riluzole attains such high potency for its cellular effects without having a high-affinity binding site.

## Results

For this work, we utilised NavMs, a prokaryotic VGSC (BacNav) from the bacterium *Magnetococcus marinus*, which we have previously demonstrated to be a good model for drug binding to eukaryotic VGSCs (eNavs)[35], and for which we have successfully determined crystal structures in apo[36,37] and drug-bound forms[37,38]. The apo-NavMs structure was initially reported to be in the open state due to its wide pore gate. However, recent work has shown this channel model to be non-conductive and also convincingly proposes that π-helix driven transitions in the channel pore (not present in the NavMs crystal structure) are required for $Na^+$ conduction, redefining this structure as representing an inactivated channel state[39]. BacNavs are homotetramers of $\sim 30\,kDa$ monomeric subunits containing 6 transmembrane helices (Fig. 1a, left), (designated S1-S6), which, when associated, form channels with the basic structure of four peripheral voltage-sensing domains (VSDs), containing helices S1-S4, each independently connected to a $Na^+$ conducting pore module (PM) in a domain-swapped arrangement that leads to the PM forming structures from one subunit packing against the VSD of a neighbouring subunit[40], (Fig. 1b). The PM houses the intramembranous $Na^+$-conducting channel and the extracellular $Na^+$ selectivity filter (SF), with the channel being formed by S5 and S6 helices from each subunit, and the SF contained within their interconnecting P-loops, which each contain selectivity filter residues sandwiched between membrane descending (P1) and ascending (P2) helices. Lateral fenestrations in the PM penetrate the channel and form intramembrane tunnels that connect the membrane core to the channel pore. This fundamental architecture is maintained in eNavs which create the channels from a single amino acid chain ($\sim 210\,kDa$), where four individual domains ($D_I$-$D_{IV}$) replace subunits to form pseudo-tetrameric structures (shown for the human Nav1.4 isoform in Fig. 1a, right and Fig. 1c).

### Riluzole binding to NavMs

To establish if riluzole binds to NavMs we used the nuclear magnetic resonance (NMR) technique Saturation Transfer Difference NMR (STD-NMR), which identifies protein-ligand interactions by reporting on internuclear saturation transfer from protein to ligand in close contact, (distances $< 7\,Å$)[41]. We used a fluorine-adapted method ($^{19}F$-$^{19}F$ STD-NMR)[42], biosynthetically labelling NavMs with 4-fluorophenylalanine to make NavMs_F-Phe. NavMs contains 16 phenylalanine residues distributed throughout the voltage sensor and pore domains common to eNavs, but none in its cytoplasmic domain found only in BacNavs. Irradiation at the NavMs_F-Phe fluorine resonance (-119 ppm) produced time-dependent saturation transfer to the riluzole trifluoro-group resonance (-58.4 ppm) when the experiment was performed in the presence of NavMs_F-Phe (Fig. 2a, Supplementary Information Fig. 1a), but not in its absence, (Supplementary Information Fig. 1b), identifying direct interaction between NavMs and riluzole.

Subsequent crystallisation trials performed with NavMs in the presence of DMSO (the riluzole vehicle solvent, 141 mM, 1%) with and without riluzole ($\sim 0.5\,mM$) yielded pyramidal-shaped crystals that diffracted to high resolution ($< 2.5\,Å$). Homotetrameric NavMs crystallised in the I422 space group with one monomer per asymmetric unit. The electron density maps for all crystals were identical except for the riluzole co-crystals (NavMs-RIL) where minor differences in the maps were consistently observed inside the NavMs fenestrations. In this region, the electron density for crystals grown in the presence of DMSO alone matched that found in the apo-NavMs map (PDB ID: 6SX5[37]) and is attributable to the hydrocarbon tails of the detergent Hega-10 which replaces lipid during purification. However, NavMs-RIL consistently had density in this region containing small additional features in their Fo-Fc difference maps, consistent with the four symmetry-related subunits producing overlapping electron density in this region reflecting partial occupancy of Hega-10 and riluzole, but with no way to deconvolute the density and unambiguously place the

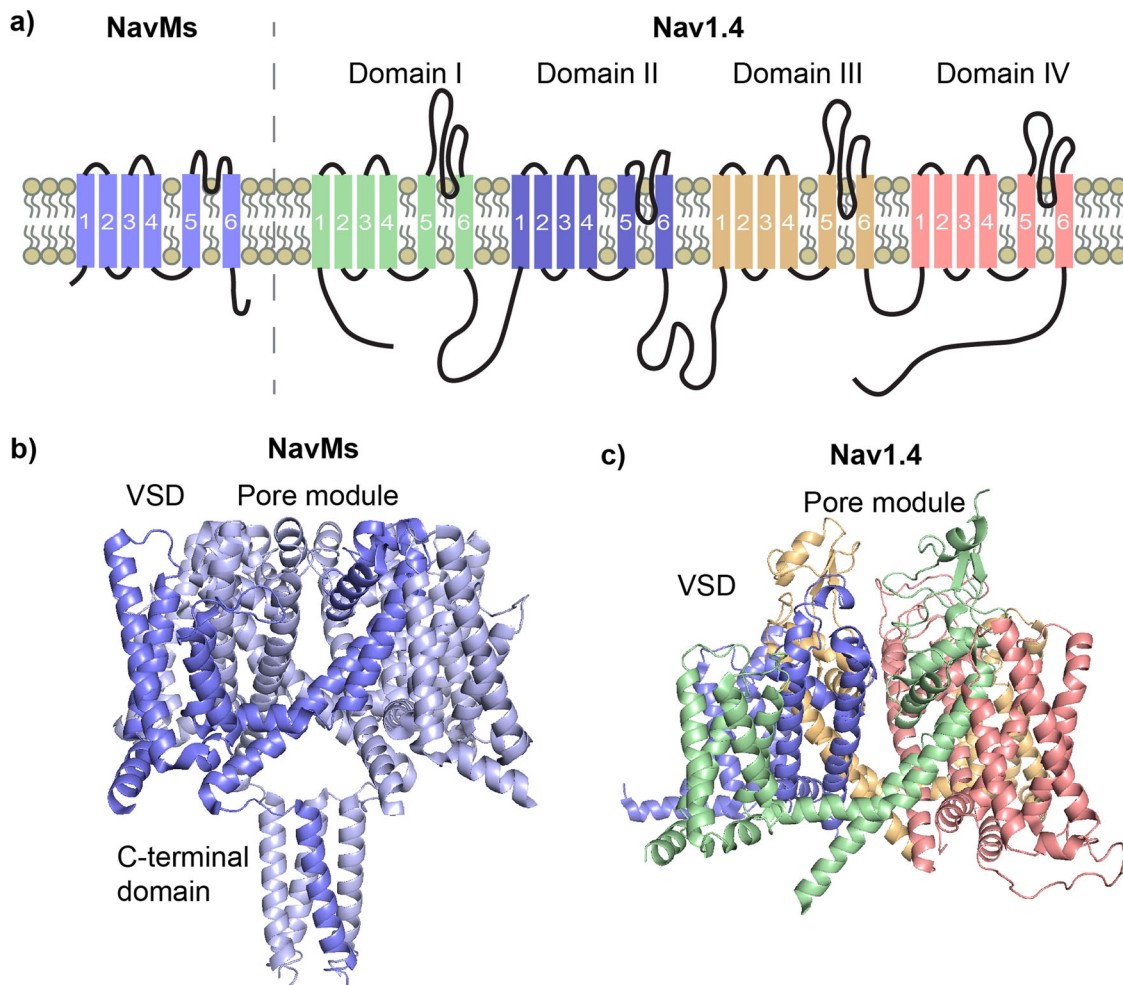

**Fig. 1 | Prokaryotic VGSCs are structurally simpler compared to eukaryotic VGSCs but share basic functional architecture. a** Basic topology in the membrane of a single chain of the prokaryotic VGSC NavMs, showing the transmembrane helices (labelled 1-6 for helices S1-S6) which forms one domain of the homotetrameric channel, (top left), next to the single chain of the eukaryotic VGSC Nav1.4 which forms all 4 domains (top right) (**b**) Four individual subunits of NavMs produce the functional homotetrameric channel in a domain-swapped arrangement (a single domain coloured in deeper blue), **c** Nav1.4 folds from the single chain to form the functional pseudo-tetrameric channel.

small (234 Da) planar riluzole molecule into this site (Supplementary Information Fig. 2a and b).

To address this, we attempted to locate the riluzole sulphur atom by collecting single-wavelength anomalous diffraction (SAD) data close (see "methods") to the sulphur absorption K-edge. Both DMSO and riluzole contain sulphur so we collected data on DMSO-only and NavMs-RIL crystals, and the anomalous density present in the X-ray data was determined. In all crystals, we successfully identified sulphur signals for 10 of the 11 methionine residues of NavMs (not M90 which is in an unstructured loop), and for its single cysteine, C52. For NavMs-RIL, an additional peak in the anomalous density map was present in the region of the extra electron density found in the fenestrations, identifying the riluzole sulphur. Collection from the first crystal (crystal 1), gave anomalous density for this peak with an intensity of ~5σ (Fig. 2b), indicative of a riluzole occupancy of ~20% relative to the strongest anomalous density peak in the data. Four separate collections, performed on a second crystal (crystal 2), each at a different crystal orientation, all produced this extra peak with the first collection producing a peak intensity comparable to crystal 1 (Supplementary Information Fig. 2c). In the subsequent collections this peak was maintained but with reduced intensities, consistent with expectations for a genuine peak while accounting for increased radiation damage impacting the signal as collections proceeded (Supplementary

Information Fig. 2d). The DMSO-only crystals contained no corresponding anomalous peak even at the level of signal noise, (Supplementary Information Fig. 2e and f). All of the crystals analysed exhibited two anomalous sulphur peaks for M204, a bottleneck residue located at the membrane entrance to the fenestration. The major peak corresponds to the open fenestrations modelled in our structure. However, the movement of the methionine sidechain to match the second peak (shown as the minor anomalous sulphur peak in Fig. 2b) closes the fenestrations (Supplementary Information Fig. 2g and h). These closed fenestrations would be empty of detergent tails in the crystals, (and in vivo, lipid tails), as they could not access the channel. Additionally, this sidechain movement would clash with the trifluoromethoxy-group of riluzole as positioned in our structure. However, an alternative riluzole orientation achieved by rotating the molecule 180° around the sulphur atom to reposition the drug with this group in the channel pore, although unlikely from the electron density maps, could be accommodated if riluzole bound to this closed fenestration state.

To eliminate any ambiguity concerning the position of riluzole in the binding site, we again performed ¹⁹F-¹⁹F STD-NMR, but this time, we used it to look for saturation transfer between NavMs, specifically fluorine-labelled at a single site in the protein, and riluzole. A trifluoromethyl group was introduced onto the cysteine of a NavMs

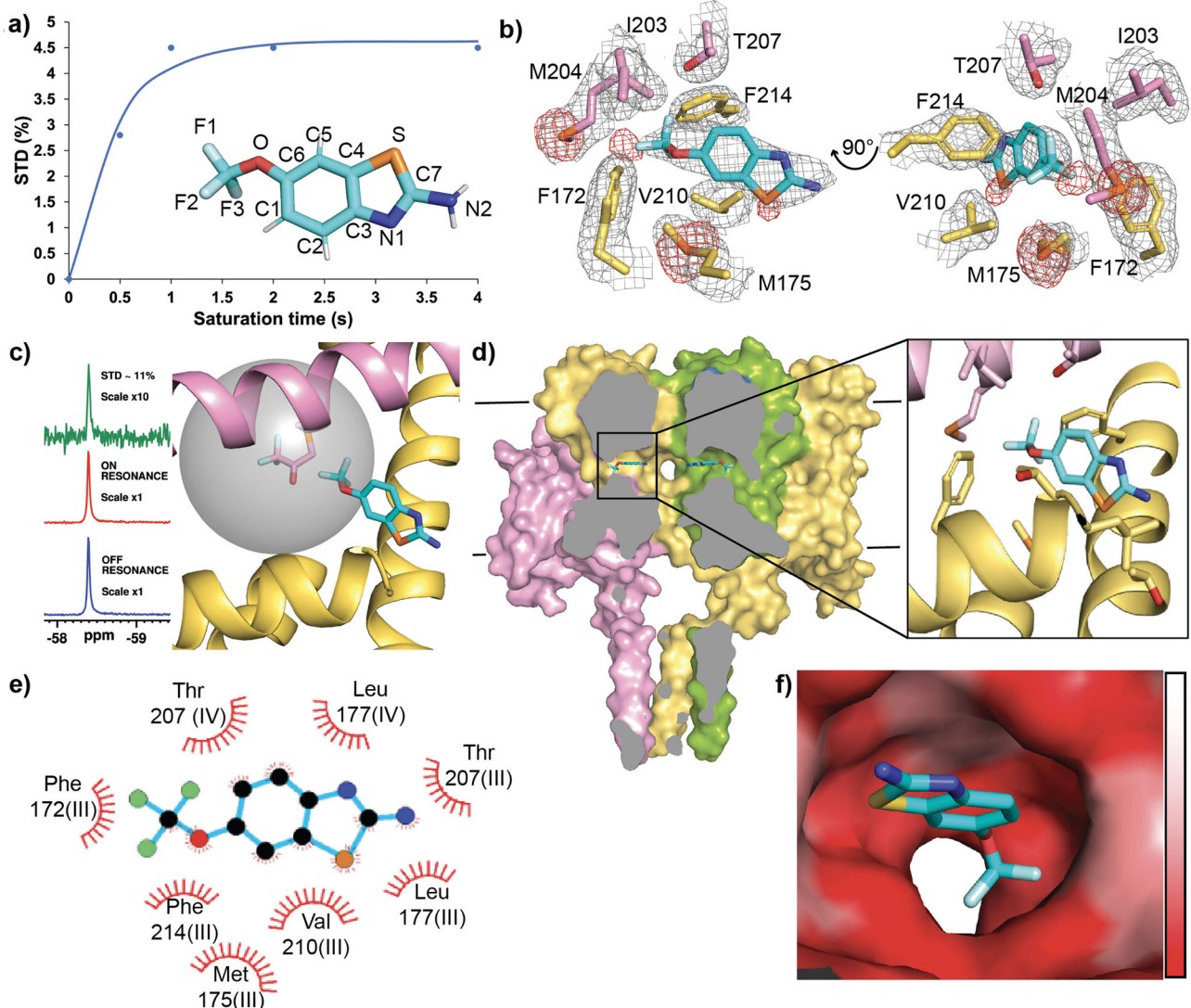

**Fig. 2 | Riluzole interaction with NavMs reveals an atypical VGSC binding site.** **a** [19]F-[19]F STD build-up over lengthening saturation times produces a curve for the interaction of riluzole with NavMs_F-Phe with an STDmax at plateau ~3.3% (inset, riluzole coloured by heteroatom, with numbering as used throughout this study). **b** Riluzole binding site in NavMs produced from x-ray crystal analysis. Two views of the rilzuole binding site in NavMs (pink and yellow denoting sidechains from interacting residues from neighbouring domains) with 2Fo-Fc map (contoured at 1σ, grey), and sulphur anomalous signals (contoured at 3.5σ, red). **c** [19]F-[19]F STD NMR using a site specific probe orientates riluzole in its binding site in NavMs. Subtraction of riluzole [19]F signal produced from irradiation at the NavMs-BTFA fluorine resonance, (ON RESONANCE left, middle) from that produced from irradiation at the control resonance, (OFF RESONANCE left, bottom) results in a large STD peak ( ~11% top, left) indicating strong saturation transfer between NavMs-BTFA and positioning the CF3-group of riluzole at the membrane side of the fenestration binding site (right panel). The grey sphere represents the radius of 7 Å which is the maximum distance for the STD effect from the fluorines on the C204-BTFA probe (shown as the modified sidechain at the centre of the sphere) (**d**) The binding site of riluzole within NavMs. (left), The complete structure of NavMs is shown cut through at fenestration depth with riluzole bound. (left,) with zoomed in view (right). **e** Ligplot analysis of the riluzole binding site with hydrophobic contacts from protein residues shown as red eyelashes. Interacting residues are labelled for their corresponding domains if transposed onto the $D_{III}$-$D_{IV}$ fenestration of eNavs (**f**) Surface view of the fenestration binding site for riluzole coloured by hydrophobicity scale (low, white to high, red) looking outward from the pore.

C52A/M204C double mutant by chemical conjugation using the alkylating agent 3-bromo-1,1,1-trifluoroacetone (BTFA), to produce NavMs-BTFA. This introduced a trifluorinated probe into NavMs at a specific location near the membrane-facing opening to the fenestrations. Selective irradiation of the [19]F resonance of NavMs-BTFA (-83.8 ppm) produced robust STD at the riluzole trifluoro-group resonance (-58.4 ppm, Fig. 2c, left), placing the fluorines of riluzole within the 7 Å radius delineated by the probe (Fig. 2c, right), and confirming the original placement of riluzole in the structure as the correct one.

Using this combination of techniques, we could unambiguously place riluzole into the binding site in the open fenestration conformation of NavMs (Fig. 2d). No movements in protein structure were observed on riluzole interaction, with riluzole binding NavMs in a hydrophobic pocket within the fenestration and making contacts with S6 helix and SF residues from two consecutive domains and residues from the P1 helix from the first of these domains (Fig. 2e, f). As a true tetramer NavMs contains four equivalent fenestrations, whereas in pseudo-tetrameric eNavs individual fenestrations are markedly different in both size and sequence[43]. The $D_{III}$-$D_{IV}$ fenestration of eNavs contains a conserved phenylalanine residue on helix $S6_{IV}$ at the pore lining of this fenestration, which forms part of the 'local anaesthetic (LA) binding site', that is key to the action of VGSC pore-blocking drugs that preferentially bind to and stabilise the inactivated channel state[44,45]. This residue has also been implicated in the eNav-riluzole

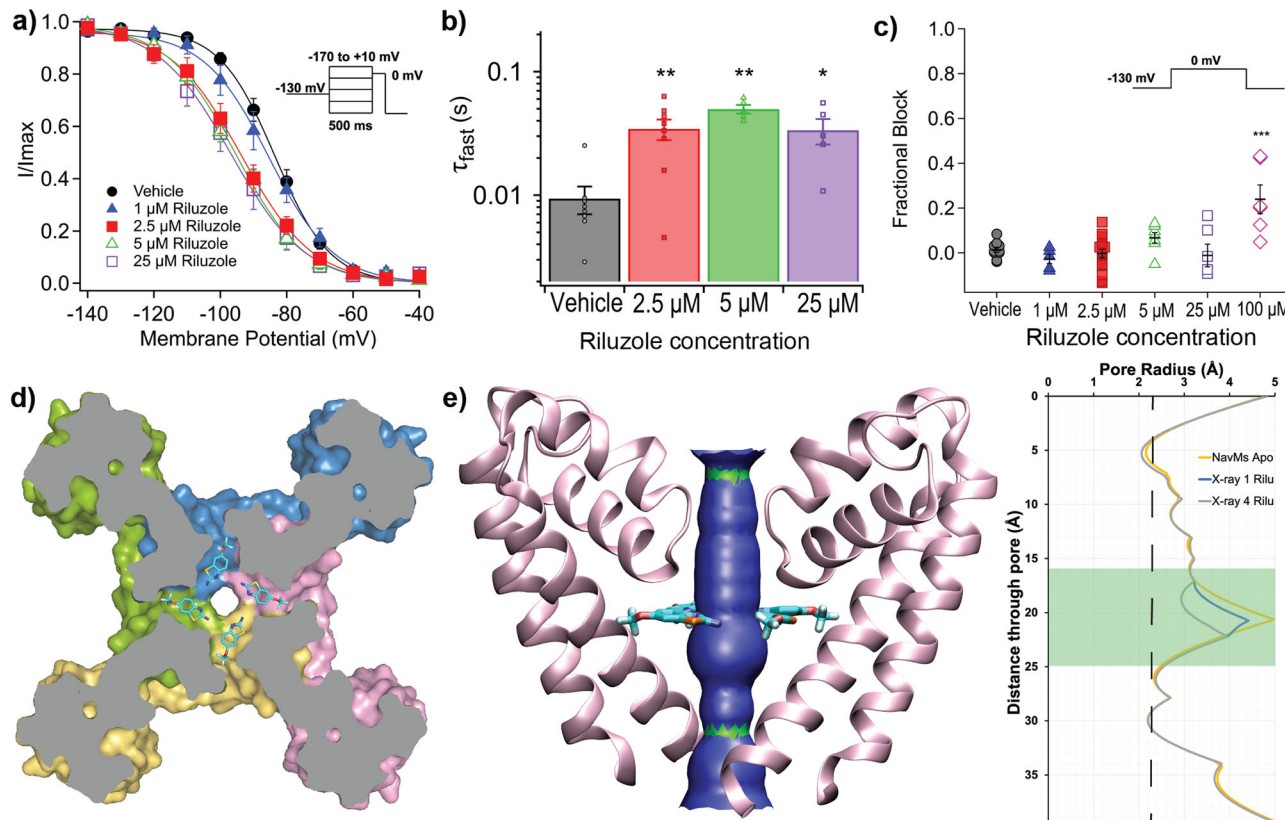

**Fig. 3 | Effects of riluzole binding eNavs are reproduced in NavMs. a** Riluzole produces a dose-dependent hyperpolarised shift in the SSI curve for NavMs. Voltage dependence of inactivation of HEK293T cells transfected with NavMS after perfusion of the vehicle (black circles) or increasing concentrations of riluzole. Data points were fit to a Boltzmann equation to calculate $V_{1/2}$ (Supplementary Information Table 2). The normalised current data is displayed as mean ± S.E.M. SSI $V_{1/2}$ after perfusion with the vehicle (black circles) was −84.5 ± 1.4 mV ($n = 11$), but was −86.9 ± 2.6 mV ($n = 5$), −94.1 ± 2.5 mV ($n = 18$), −95.5 ± 1.8 mV ($n = 8$), and −97.3 ± 3.2 mV ($n = 6$) after perfusion with 1 μM (blue triangles), 2.5 μM (red squares), 5 μM (green triangles), and 25 μM (purple squares) riluzole respectively. One-way ANOVA and post-hoc Tukey test found that 2.5, 5 and 25 μM riluzole significantly shifted SSI compared to vehicle ($p = 0.025$, 0.038, and 0.024 respectively). **b** Riluzole slows NavMs recovery from inactivation in HEK293T cells. The $t_{fast}$ component of a biphasic recovery process was significantly increased by 3-fold. Data displayed as mean ± S.E.M, *$p < 0.05$, **$p < 0.01$. $τ_{fast}$ after perfusion with the vehicle was 0.009 ± 0.002 s ($n = 8$), but was 0.034 ± 0.007 s ($n = 8$), 0.050 ± 0.004 s ($n = 5$), and 0.033 ± 0.008 s ($n = 6$) after perfusion with 2.5, 5 and 25 μM riluzole respectively. One-way ANOVA and post-hoc Tukey test found that that 2.5, 5 and

25 μM riluzole significantly shifted $τ_{fast}$ compared to vehicle ($p = 0.008$, 0.0002, and 0.027 respectively). **c** Riluzole only blocks NavMs in HEK293T cells at the high concentration (100 μM) with 1–25 μM producing no significant occlusion of the channel pore. Data points shown with mean ± S.E.M indicated, ***$p < 0.0001$. Fractional block after perfusion with the vehicle was 0.002 ± 0.016 ($n = 11$), and was −0.027 ± 0.021 ($n = 5$), −0.007 ± 0.024 ($n = 18$), 0.055 ± 0.068 ($n = 8$), −0.011 ± 0.050 ($n = 6$), and 0.239 ± 0.064 ($n = 6$) for 1 μM (blue triangles), 2.5 μM (red squares), 5 μM (green triangles), 25 μM (purple squares), and 100 μM (pink diamond) riluzole respectively. One-way ANOVA and post-hoc Tukey test found that only 100 μM riluzole significantly blocked NavMS compared to vehicle ($p = 0.0001$). **d** Top-down sliced view of NavMs at fenestration depth showing riluzole bound in all four fenestrations. **e** Side view from Hole2 analysis showing that even with all 4 fenestrations occupied, riluzole does not block Na⁺ conduction. A pore radius of > 2.3 Å is required for Na+ conduction and is represented by blue in the channel tunnel cartoon (left panel) and delineated by the broken vertical line in the plot of pore radius along the pore axis (right panel). The green shaded area in the plot (right panel) represents Na⁺ conduction pathway at fenestration depth.

interaction[19]. The riluzole binding site in NavMs includes T207 from the S6 helix of the second binding domain which, if this fenestration was transposed onto the $D_{III}$-$D_{IV}$ fenestration of eNavs, represents the position of this $S6_{IV}$ residue, ($D_{III}$-$D_{IV}$ equivalent labelling added to NavMs interacting residues in Fig. 2e). Additionally, the P1 helix and SF of eNavs, both containing residues forming part of the riluzole binding site in NavMs, are also involved in inactivation[46]. As a model for eukaryotic inactivation, BacNavs only have slow inactivation processes reminiscent of slow inactivation in eNavs. However, it has been shown that riluzole modulates both fast and slow inactivation of eNavs with equivalent potency[47], indicative of a common binding event inhibiting both processes.

## NavMs as a functional model for riluzole action on eNavs
At therapeutically relevant concentrations it has been shown that riluzole selectively stabilises eNavs in their inactivated state, which in

whole-cell studies manifests as riluzole causing dose-dependent hyperpolarisation of steady-state inactivation (SSI) and slowed recovery from inactivation (RFI), without it affecting activation kinetics or producing channel block[9,10]. We performed whole-cell patch clamp experiments using Human embryonic kidney 293 T (HEK293T) cells transiently transfected to express NavMs and found that this profile of riluzole effects on eNavs was replicated on NavMs (Supplementary Information Tables 2 and 3). Riluzole dose-dependently shifted SSI of these cells in the hyperpolarised direction (Fig. 3a), with the IC₅₀ of 2 μM for the shift (Supplementary Information Fig. 3a) being comparable to the IC50 values for similar shifts of SSI found with eNavs[47]. Riluzole slowed the RFI of NavMs, with 2.5 μM riluzole causing a 3-fold greater delay in channel recovery compared to vehicle (Fig. 3b). Activation kinetics were unaffected by riluzole, with no change in steady-state activation or voltage-dependence of activation of NavMs up to 25 μM, (Supplementary Information Fig. 3b and c), and channel block

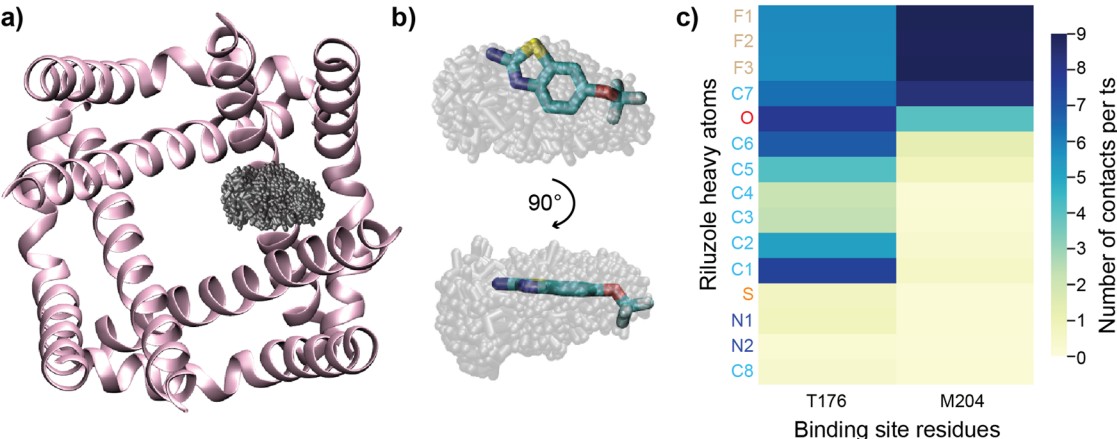

**Fig. 4 | MD simulation of riluzole with NavMs. a** Clustering analysis of riluzole and NavMs, all members of the primary cluster (depicted in grey) and NavMs WT (depicted as pink helices in a top-down view at the depth of the fenestrations. The primary cluster represents the riluzole location for 26% of the total simulation time, **b** The primary cluster overlays with the binding location of riluzole in the X-ray structure. **c** Contact map of riluzole atoms with T176 and M204 from NavMs calculated from the simulation, coloured by the average number of contacts per timestep, showing that riluzole orientation is the same in both structurally-determined and MD simulation-determined binding sites.

was absent up to the same concentration (Fig. 3c, Supplementary Information Fig. 3d, e), although, similarly to eNavs[9,47], some block developed at high concentration (Fig. 3c, 100 μM). A riluzole binding site in NavMs that can produce the effects of the drug at therapeutically relevant concentrations is consistent with its location in the crystal structure (Fig. 3d) where binding does not occlude the Na$^+$ conduction pathway, even if all four fenestrations are occupied (Fig. 3e). The channel block produced at high concentration results from riluzole either binding to a secondary binding site located in the pore or through it non-specifically occluding of the pore, with either scenario being functionally-irrelevant at therapeutic riluzole concentrations.

## Molecular dynamics simulations reveal the mechanism of riluzole interaction

To investigate how riluzole interacts with NavMs in a membrane environment we performed a 2 μs unbiased molecular dynamics (MD) flooding simulation with NavMs embedded in a POPC lipid bilayer and riluzole molecules initially distributed in the aqueous phase. Due to the high concentration of drugs required in flooding experiments, combined with riluzole hydrophobicity, some riluzole molecules non-specifically aggregated on the aqueous-exposed surfaces of NavMs at the membrane interface at the beginning of this simulation and these were excluded from subsequent analysis. During the 2 μs simulation, lipid tails quickly and dynamically occupied the fenestrations of NavMs and riluzole molecules rapidly partitioned into the lipid bilayer. The simulation, when assessed for contacts between any riluzole molecule and NavMs, found binding hotspots only in the NavMs fenestrations (Supplementary Movie 1). Spatial clustering was carried out over the length of the simulation for a typical binding riluzole molecule using an RMSD threshold of 2 Å, producing a primary cluster occupied by riluzole for 26% of the total simulation time (Fig. 4a). Superimposing this cluster onto the NavMs-RIL crystal structure showed that riluzole would be entirely nested within this cluster (Fig. 4b).

To understand the molecular interaction between NavMs and riluzole, distance-based intermolecular contacts were calculated to quantify the number and nature of the interactions within the binding site throughout the simulation. A contact was defined as when any atom of a riluzole molecule was within 6 Å of both the M204 residue, at the membrane-facing entrance to the fenestration, and the T176 residue, which is closer to the channel pore. This analysis allowed for a maximum residence time for riluzole binding to be determined to

quantify the strength of the interaction, and preferred orientations to be calculated. Riluzole bound NavMs at this site with a maximum residence time of 92 ns, with the contact map produced for the interaction (Fig. 4c) showing that riluzole molecules bound to NavMs in the same orientation and with similar atomic contacts as those of the crystal structure.

## Importance of the Local Anaesthetic (LA) binding site residue for riluzole action on VGSCs

Riluzole (100 μM) effectiveness in stabilising inactivation is diminished in an eNav (rat Nav1.4) by alanine substitution of the LA binding site phenylalanine residue[19]. Using whole-cell patch clamp experiments, we investigated the importance of this residue on VGSCs at clinically relevant riluzole concentrations, initially comparing how riluzole-affected HEK293T cells transiently transfected to express WT human Nav1.4 (hNav1.4) to those expressing the LA binding site mutant channel hNav1.4 F1586A. Riluzole (1 μM) stabilised the inactivation of the WT channel, producing a hyperpolarised shift in the SSI curve (Supplementary Information Fig. 4a). The same experiment produced no shift when WT hNav1.4 was replaced with the F1586A mutant (Supplementary Information Fig. 4c) showing that this residue was also vital for riluzole's stabilisation of the inactivation of hNav1.4 at this low concentration. Consistent with other studies, riluzole did not affect the activation of either channel type (Supplementary Information Figs. 4b and 4d). To investigate this site for the riluzole effect on inactivation on NavMs, a similar patch-clamp experiment was performed as previously, but with cells expressing NavMs T207A. The addition of 2.5 μM riluzole, which had produced a ~10 mV hyperpolarised shift in the SSI curve with the WT channel as characterised by the $V_{1/2}$ for the shift ($V_{1/2}$ = −84.5 ± 1.4 mV vehicle, −94.1 ± 2.5 mV riluzole), failed to produce a significant shift on T207A ($V_{1/2}$ = −84.7 ± 2.6 mV vehicle, −88.6 ± 1.6 mV riluzole) (Fig. 5a, Supplementary Information Table 3), showing that this site is also crucial for riluzole to stabilise the inactivated state of NavMs, even though the residue is a threonine and not a phenylalanine.

To investigate how this mutation affects NavMs on a structural level, we ran an analogous MD simulation to that performed previously, replacing WT NavMs with T207A. In this simulation, riluzole entered both membrane and fenestration similarly to the WT simulation. However, once in the fenestration, sustained binding was not seen, with riluzole molecules continuing to be mobile in the pore. This simulation produced no majority spatial cluster (ie., no cluster > 5% occupancy). Consequently, riluzole contacts at the binding site were

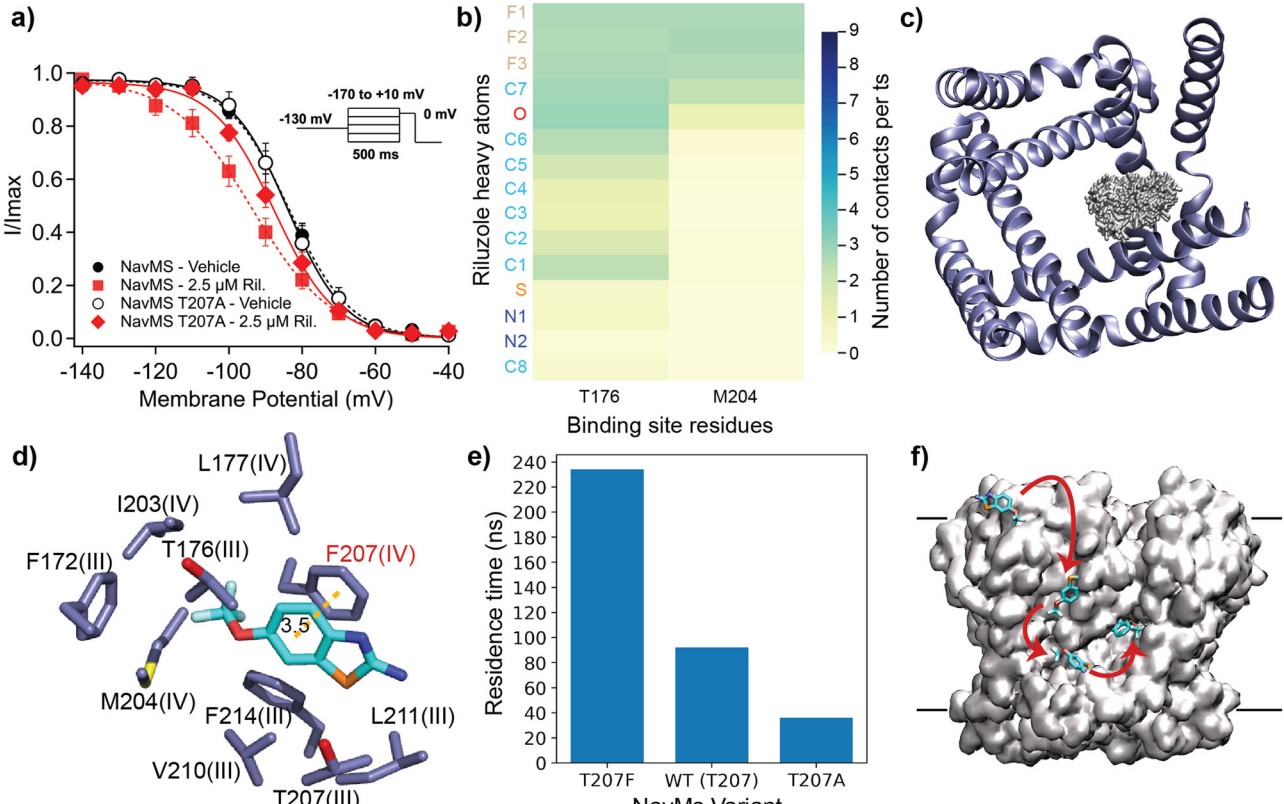

**Fig. 5 | Importance of the LA binding site for riluzole binding and action in NavMs and the overall mechanism of interaction. a** LA binding site mutation T207A abrogates the riluzole effect on SSI of NavMs expressed in HEK293T cells. Data points for WT NavMs (circles) and NavMS T207A (squares) were fit to a Boltzmann equation to calculate $V_{1/2}$ (Supplementary Information Table 2). The normalised current data is displayed as mean ± S.E.M. Data points were fit to a Boltzmann function to calculate $V_{1/2}$. Unpaired Student's t-test of the riluzole effect on SSI of NavMs T207A found no significant difference compared to NavMs T207A without riluzole ($n = 5$, $p > 0.5$), **b** Contact map for riluzole interaction with NavMs T207A over the course of the simulation showing that the average number of

riluzole atoms which are in simultaneous contact with residues T176 and M204 are greatly reduced compared to WT **c** Top-down view at fenestration depth of all members of the primary cluster (grey) from the NavMs T207F MD simulation. **d** Riluzole binding site within this primary cluster represents riluzole binding in the eNav $D_{III\text{-}IV}$-mimicking fenestration of NavMs T207F with the additional π–π stacking interaction indicated by a broken yellow line **e** Bar chart of the maximum riluzole residence times observed in the binding sites of each NavMs variant (**f**) Pathway taken by a typical NavMs binding riluzole molecule during the MD simulations. Black lines represent the membrane boundary.

much fewer (Fig. 5b), resulting in a greatly reduced maximum residence time for T207A (36 ns, Supplementary Movie 2) compared to WT (Fig. 5e, compare Supplementary Movies 1 (WT) and 2 (T207A)), showing that riluzole and NavMs interactions were less frequent and weaker compared to WT.

Considering the importance of the LA binding site for the action of riluzole, we used the same MD simulation to explore the impact of introducing a 'humanising' T207F mutation into one subunit of the NavMs tetramer. In this simulation, there was a clear preference for binding in this eNav $D_{III}$-$D_{IV}$ interface-mimicking fenestration compared to the native fenestrations, resulting in it containing the primary cluster, representing 32% occupancy of simulation time (Fig. 5c). This cluster produced a binding site location slightly closer to the pore compared to WT, but riluzole bound in the same orientation and maintained a non-pore-blocking pose (Supplementary Information Fig. 5a and b). The introduction of F207 resulted in a binding interaction where the benzothiazole ring of riluzole π–π stacked with F207 (Fig. 5d, Supplementary Movie 3), stabilising the interaction, and resulting in riluzole having a higher maximum residence time in this fenestration (234 ns) compared to WT (Fig. 5e).

**A hydrophobic pathway links VGSC binding to riluzole potency**
The MD simulations with NavMs reveal the basis for how riluzole can potently affect VGSC function. In these simulations, riluzole rapidly

partitioned out of the aqueous phase to form a highly concentrated repository of riluzole in the membrane (all riluzole molecules entered the membrane within 100 ns in a membrane-only simulation). Therefore, riluzole action is not reflective of its measured aqueous concentration but results from a much higher concentration in the membrane. Membrane partitioning was facilitated by the highly lipophilic trifluoromethyl group of riluzole. Once in the membrane riluzole (which often non-specifically sampled NavMs) enters the fenestration, amine group first, to access its binding site, (complete pathway taken by a typical riluzole molecule shown in Fig. 5f and dynamically in (Supplementary Movie 4).

**MD simulations with riluzole and human Nav1.4**
Eukaryotic VGSCs are incompatible with the structural techniques used in this study, and cryoEM has only revealed a pore-blocking pose for the drug[48], consistent with a second binding site causing the channel block produced by riluzole at high non-therapeutic concentrations. However, considering the accuracy of MD simulations in reproducing the interaction of riluzole with NavMs found in our structural study, we sought to use it to investigate the effect of riluzole on a human VGSC (hNav). We used the same MD simulation as before but replaced NavMs with hNav1.4 (the first available cryoEM structure for an hNav, PDB ID: 6AGF[49], also captured in the inactivated state). Here, after ~500 ns, riluzole bound in the $D_{IV}$-$D_I$ fenestration where it

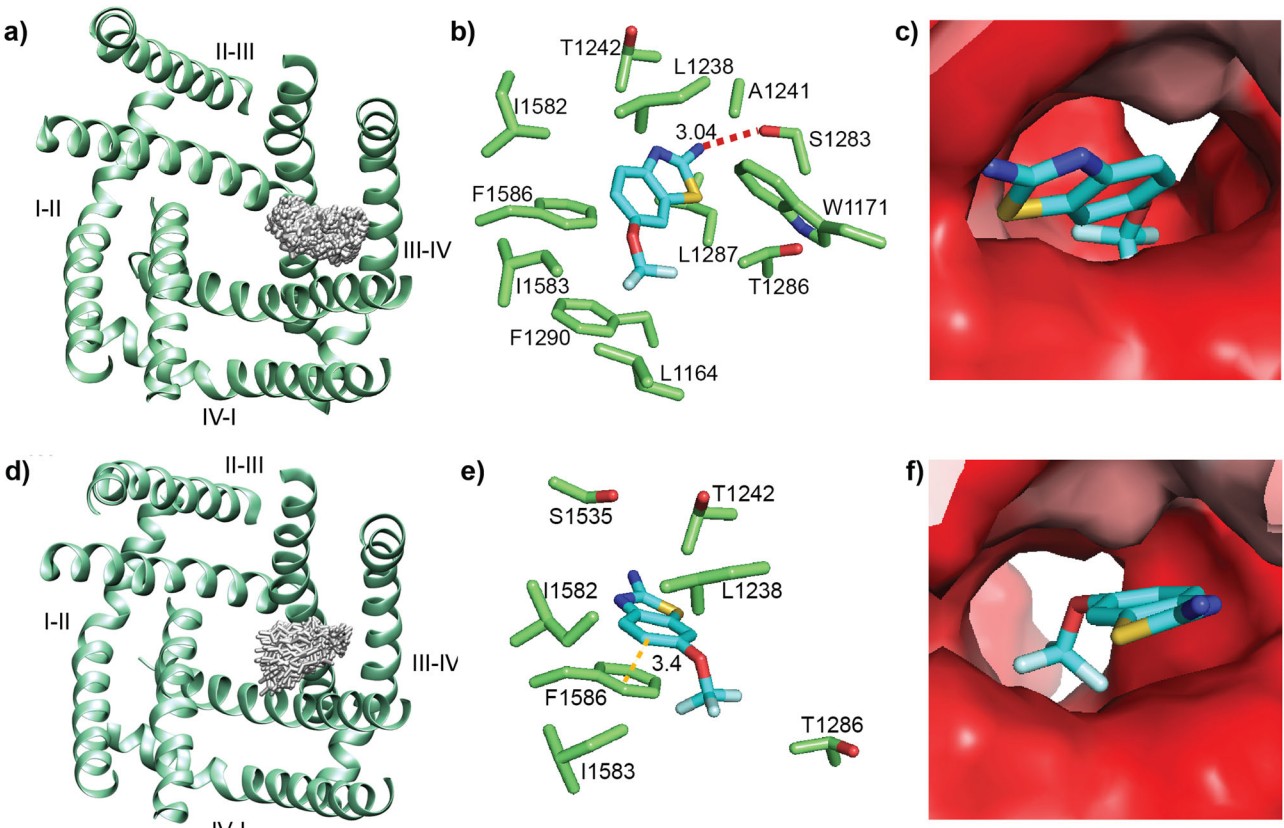

**Fig. 6 | MD simulation shows two riluzole binding sites in the hNav1.4 $D_{III}$-$D_{IV}$ fenestration. a** Cluster 1 (grey) for riluzole interaction in the hNav1.4 $D_{III}$-$D_{IV}$ fenestration. **b** Interacting residues at binding site 1 showing a hydrogen bond between riluzole and S1283. **c** Surface view of riluzole bound in the fenestration at binding site 1 looking outward from the pore. **d** Cluster 2 (grey) for riluzole interaction. **e** Interacting residues at binding site 2 features a π–π stacking interaction between the benzothiazole group of riluzole and F1586. **f** Surface view of riluzole in the fenestration at binding site 2 looking outward from the pore. Figure 6c, f surfaces are coloured for hydrophobicity on the scale shown in Fig. 2f.

remained for the rest of the 2μs simulation. Riluzole bound in a cluster further away from the channel pore compared to the NavMs simulations (Supplementary Information Fig. 6a), binding in a hydrophobic pocket, with π–π staking between the benzothiazole group of riluzole and F432 of hNav1.4 (Supplementary Information Fig. 6b). However, this interaction does not involve the $S6_{IV}$ residue F1586 which we have shown to be crucial to the effect of riluzole on hNav.1.4 and therefore lacks relevance for any known therapeutic effect of the drug. In this simulation, lipids extended deeper into the $D_{III}$-$D_{IV}$ fenestration compared to that of $D_{IV}$-$D_I$, visibly interacting with the pore-lining F1586 residue, making riluzole interaction less likely to occur in this fenestration during the 2μs of simulation.

To alleviate this constraint, we performed a slightly modified MD simulation, placing a riluzole molecule at the entrance to the $D_{III}$-$D_{IV}$ fenestration at the beginning of the simulation, allowing it to compete with lipids for binding from the start. In this simulation, riluzole immediately moved into the fenestration (within the first nanosecond), where it stayed for the remainder of the simulation. This simulation produced two major riluzole binding clusters, both contained within overlapping hydrophobic pockets inside the $D_{III}$-$D_{IV}$ fenestration, with both involving interaction with F1586. Comparably with NavMs, these clusters produced binding sites that included residues from the S6 helices, P1 helix, and SF contained within the fenestration (Supplementary Information Fig. 7). Cluster 1 (Fig. 6a, c) sits centrally to the fenestration and produces a binding site almost identical in its alignment to that of the WT NavMs (Supplementary Information Fig. 7, top alignment) Here, riluzole binds almost exclusively through hydrophobic interactions, although a weak hydrogen

bond exists between the amine of riluzole and the side chain hydroxyl of S1283 on $S6_{III}$ (Fig. 6b). Cluster 2 (Fig. 6d, f) lies slightly closer to the pore of the channel and produces a binding site almost identical in alignment to that produced in the simulation of NavMs T207F (Supplementary Information Fig. 7, bottom alignment) where riluzole binds exclusively by hydrophobic interactions and features a π–π stacking interaction between the benzothiazole ring of riluzole and F1586 (Fig. 6e).

To investigate the importance of F1586 to the interaction, we performed the same simulation with hNav1.4 F1586A. Here, riluzole again entered the fenestration but, similarly to the simulation involving NavMs T207A, showed higher mobility, lower occupancy, and greatly reduced binding site contacts relative to WT (Supplementary Information Fig. 6c, d), consistent with our functional experiments showing the diminished effect produced by riluzole on the F1586A channel compared to WT.

## Riluzole normalises the pathological $I_{NaL}$ produced by a Nav1.4 disease variant

Humans have nine VGSC subtypes differentially expressed across excitable tissues of the body. The hNav1.4 used in this study is a VGSC isoform expressed predominantly in skeletal muscle, and mutation within this channel is causative to many muscular diseases, including myotonias and/or periodic paralyses[11,50]. Myotonias are hyperexcitable states that are associated with altered inactivation kinetics and elevated $I_{NaL}$[11,50], and it has been reported that riluzole has high potency against cells expressing WT hNav1.4 when mimicking a hyperexcitable myotonia-like state, (IC50 ~0.9μM), compared to normal states,

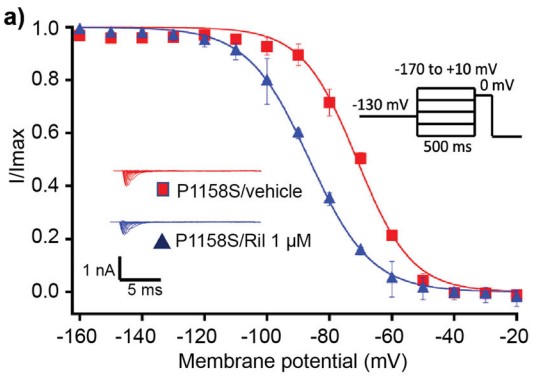

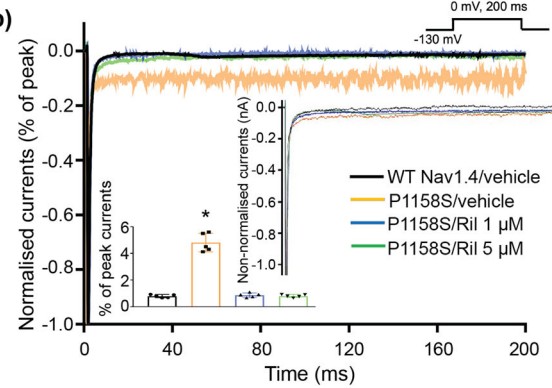

**Fig. 7 | Riluzole stabilises inactivation in a hNav1.4 myotonia mutant. a** Effect of riluzole (1 μM) or its vehicle solvent (DMSO) on the SSFI of hNav1.4 P1158S expressed in HEK293T cells. Data is plotted as average (± S.E.M) with the insert showing the protocol and representative currents ($n = 5$, each). Data points were fit to a Boltzmann function. **b** Representative $I_{NaL}$ showing the effect of riluzole (1 or 5 μM) or its vehicle solvent on hNav1.4 P1158S expressed in HEK293T cells. The inset bar graph shows the effect of riluzole on the percentage (± S.E.M) of $I_{NaL}$ ($n = 5$, each). One-way ANOVA and post-hoc Tukey test found that hNav1.4 P1158S significantly increased $I_{NaL}$ ($p < 0.0001$) compared to WT, and that application of 1 and 5 μM riluzole significantly decreased $I_{NaL}$ ($p < 0.0001$ for both). $I_{NaL}$ levels produced on addition of 1 and 5 μM riluzole to hNav1.4 P1158S were not significantly different from WT vehicle ($p > 0.9$ for both).

(IC50 ~50 μM)[33]. We have previously shown that the inactivation-impaired hNav1.4 variant P1158S[51], which causes myotonia and periodic paralysis, has a pathologically increased $I_{NaL}$ (~5% of peak), compared to the WT channel (~0.75% of peak)[52]. The addition of 1 μM riluzole to HEK293T cells transiently transfected to express hNav1.4 P1158S stabilised the inactivation in this variant, significantly shifting the SSI curve to more negative potentials, compared to vehicle (Fig. 7a, $p < 0.03$), and restored elevated $I_{NaL}$ to WT levels (Fig. 7b). No significant additional change to $I_{NaL}$ occurred on the addition of 5 μM riluzole (Fig. 7b).

## Discussion

This study reveals an atypical binding mechanism for a VGSC-drug interaction that is utilised by riluzole to produce effects consistent with its use at therapeutically relevant concentrations in models of ALS and other hyperexcitability-associated diseases. Riluzole follows the hydrophobic membrane accessible pathway thought to be taken by many other small neutral VGSC drug forms which are believed to pass through the fenestrations of these channels to bind in, and then block, the channel pore[53,54]. However, unlike these drugs, riluzole does not pass through fenestrations but binds within them, with binding at a location which allows the direct interaction between riluzole and a conserved residue on S6$_{IV}$ (F1586 in the D$_{III}$-D$_{IV}$ fenestration of hNav1.4 or T207 at the equivalent S6 position of NavMs) critical for the observable action of the drug. This binding event results in riluzole selectively stabilising the inactivated state of the channel without occluding the Na$^+$ conduction pathway, and manifests in cells as riluzole inhibiting $I_{NaL}$ which leads to the reduced excitability of hyperexcitable cells without it affecting $I_{NaT}$ which is vital for the normal function of the cell. The pursuit of therapeutics which target a selective channel property without binding in the channel pore is the goal of numerous modern VGSC drug development programmes. Still, as of now, few have even made it as far as clinical trials[21]. However, here we show how riluzole, a well-tolerated FDA-approved drug in clinical use since the 1990s, acts in this way.

Binding in the fenestrations of voltage-gated ion channels has already been demonstrated for drugs which produce allosteric effects on potassium (e.g., retigabine[55]) and calcium (e.g., nifedipine[56]) channels, and there is a growing list of other channels, GPCRs and transporters where membrane access pathways lead to allosteric intramembrane binding sites for drugs that modulate their targets[57]. These interactions involve drugs that localise into the membrane while potency is usually calculated from the concentrations of these drugs in

the aqueous phase. Therefore, high affinity binding to achieve high potency is not necessarily required due to the high local concentrations of the drug around these intramembrane target binding sites[57,58]. This could account for riluzole being considered a well-tolerated drug, as the low concentration of the drug outside the membrane combined with the absence of any identified high-affinity extracellular or intracellular riluzole binding targets would limit side effects.

Here we show that riluzole is the first example of a clinically employed drug that uses a membrane-access pathway to allosterically modulate a VGSC, in this case, resulting from riluzole binding within the intramembrane fenestrations of VGSCs to produce the potent and selective inhibition of $I_{NaL}$ over $I_{NaT}$. That this interaction can lead to the normalisation of the pathologically elevated $I_{NaL}$ demonstrated in models of ALS[15,16] and here, with a hNav1.4 disease variant, suggests that riluzole can preferentially target hyperexcitable disease states over normal ones. This highlights the attractiveness of similarly acting drugs for future drug design and supports the investigation of riluzole repurposing for other diseases where $I_{NaL}$ dysfunction is implicated as a cause.

## Methods

### Molecular biology

Site-directed mutagenesis of the NavMs DNA sequence contained within the pET-15b plasmid vector was performed to create the NavMs C52A M204C doubly mutated channel and NavMs T207A using Quikchange site-directed mutagenesis (Stratagene) and the primers listed in Supplementary Information Table 4. Plasmids generated were chemically transformed into NEB® 5-alpha competent E.coli (New England Biolabs) and mutation verified by automated DNA sequencing (Eurofins Genomics), before transformation of correct plasmids into Overexpress™ C41(DE3) Chemically Competent E.coli cells (Lucigen) for protein expression.

### Protein expression and purification

NavMs proteins were expressed in C41(DE3) E.coli cells grown in Luria Broth containing 100ug/ml ampicillin, in shaking flasks at 37°C. To induce NavMs expression, when $A_{600}$ reached an optical density of 0.8, IPTG was added (0.5 mM) followed by further incubation for 3.5 h. Cell pellets were resuspended in 20 mM Tris pH7.5, 150 mM NaCl, 1 mM MgCl$_2$ and cells broken by pressure. Cell debris was removed by centrifugation at 20,000xg for 30 mins and the resulting supernatant was spun at 195,000xg for 2 hours to pellet membranes. Protein was extracted from the membranes using a buffer containing 20 mM Tris,

pH 7.5, 300 mM NaCl, 20 mM imidazole and 1.5% DDM (Anatrace), on a rotating shaker at 4°C for 2 hours. Solubilised proteins were loaded onto a 1 ml HisTrap HP column (Cytiva Life Sciences) which was washed, and detergent exchanged, on the column with buffer containing 20 mM Tris–HCl, pH 7.5, 300 mM NaCl, 50 mM imidazole and 0.52% HEGA-10 (Anatrace), prior to protein elution using the same buffer with 1 M imidazole. The eluate volume was reduced (and imidazole concentration lowered to ~50 mM by sequential adding of buffer without imidazole), using a 100 K cut-off concentrator. NavMs was treated with thrombin protease (Merck) for 16 hours at 4 °C then purified by size exclusion chromatography using a Superdex200 increase 10/300 column (Cytiva Life Sciences). NavMs was concentrated to 10 mg ml⁻¹ and either used directly or stored at −80 °C. Point mutations were generated by Quikchange site-directed mutagenesis (Agilent).

Expression of NavMs_F-Phe followed the same basic protocol, but here M9 minimal media[59] was used for growth and 0.5 g/l of 4-fluorophenylalanine (Merck) was added to the cultures 15 minutes before IPTG induction.

To avoid confusion with the other mutants made in this study we refer to $NavMs_{F208L}$ as WT because in previous studies they have proved to be functionally indistinguishable[37,38]. However, $NavMs_{F208L}$ produces crystals that diffract to slightly higher resolutions making them more amenable to crystallographic study.

**BTFA-labelling.** Following purification, NavMs C52A/M204C protein was diluted to 20uM. 10 mM BTFA (Merck) was added, and the mixture was incubated at 4 °C for 24 hours with gentle rotation. Free BTFA was removed from protein by gel filtration using a Superdex200 increase 10/300 column and gave a protein peak identical in elution volume when compared to untreated protein. Complete removal of the probe was confirmed from the 1D ¹⁹F NMR spectrum of the NavMs-BTFA protein sample.

## Structural techniques

**Saturation Transfer difference Nuclear Magnetic Resonance.** Experiments were performed with 0.5 mM riluzole in buffer (20 mM Tris-Cl, 300 mM NaCl, 0.52% Hega-10) +/- fluorinated NavMs protein (50uM, monomeric protein concentration, 12.5uM tetramer).

¹⁹F-¹⁹F STD NMR experiments were performed on a Bruker Biospin Avance IIIHD 700 spectrometer, equipped with a 5 mm TCI cryoprobe operating at a Larmor Frequency of 658.66 MHz, using the acquisition software Topspin 3.5. For ¹⁹F STD NMR spectra were acquired by collecting alternating on- and off-resonance ¹⁹F spectra with saturation being achieved using a train of Gaussian-shaped pulses of duration 50 milliseconds with a peak B1 field of 500 Hz. Intermolecular ¹⁹F saturation transfer from NavMs ¹⁹F-Phe labelled (-119 ppm) or BTFA-labelled NavMs (-83.8 ppm) resonances to the riluzole resonance (-58.4 ppm) were determined over randomised collections at time points from 0 up to 4secs. Data were processed in Topspin 3.5 (Bruker) and fitted to the mono-exponential equation:

$$STD(\%) = STD_{max}[(1 - \exp(-k_{sat}t)] \qquad (1)$$

where $STD(\%) = (V_{off} - V_{on})/V_{off} \times 100$ where $V_{off}$ and $V_{on}$ are the peak integrals from the off and on resonance saturation transfer NMR spectra. $STD_{max}$ represents the STD at the maximal plateaued value.

**Crystallography.** Before crystallisation, NavMs (10 mg/ml, 300uM) was mixed with either 0.5 mM riluzole (from 50 mM riluzole stock in 100% DMSO, 1% final DMSO concentration), or 1% DMSO (141 mM) alone, and incubated for 16 hours at 4°C before setting up crystal trays. Crystallisation proceeded in sitting drops at 4°C with drops containing 75 nl well condition and 75 nl NavMs +/- DMSO/riluzole dispensed using a mosquito® crystal robot (SPT Labtech). The best crystals were grown in the well conditions containing 30-35% PEG 300, 100 mM HEPES (pH7) and 0-100 mM NaCl. Crystals were harvested and flash-frozen in liquid nitrogen directly from drops.

**Data Collection and Processing.** Crystal optimisation was required in this study and X-ray diffraction data was collected for crystals using the beamlines P13 (EMBL, Hamburg DE), ID23-1 and ID30A-1 (ESRF, Grenoble Fr) and I24 and I04 (Diamond Light Source, Oxford UK) working at wavelengths between 0.9537 and 0,9795 Å. Indexing and integration were performed with the XIA2-DIALS pipeline[60,61], followed by scaling and merging with AIMLESS[62]. Molecular replacement was performed using PHASER[63] using the search model (PDB ID: 6SX5) followed by model building in COOT[64] and structure refinement using REFMAC[65] and BUSTER[66]. Graphic illustrations were produced with PYMOL[67].

**Long Wavelength X-ray Crystallography.** Data were collected on the long-wavelength MX beamline I23 at Diamond Light Source. Data used in this study was collected at the wavelength of 2.755 Å (4.5 keV). This wavelength, although not optimal for sulphur (K-edge 5.01 Å, 2.47 keV), was selected as a compromise between the increasing signal from sulphur as it reaches its K-edge and the decreasing data quality that results from increasing X-ray absorption at longer wavelengths[68]. Initial data collected near the K-edge, at a wavelength of 4.5 Å (2.75 keV), produced anomalous sulphur signals of much lower intensity (<50% signal levels compared to data collection at 2.755 Å, 4.5 keV).

Each dataset consisted of 360° rotation with an exposure time of 0.1 s per 0.1° image. Multiple datasets of crystal 2 were taken at varying kappa and phi values to ensure the fidelity of the anomalous signals. Crystallographic data processing was performed as above. Anomalous difference maps were generated using the program Anode[69].

## Electrophysiology

Human embryonic kidney 293 T cells (HEK293T, ATCC CRL-1573) were grown at pH 7.4 in filtered sterile DMEM nutrient medium (Life Technologies, Thermo Fisher Scientific, Waltham, MA, USA), supplemented with 5% FBS and maintained in a humidified environment at 37 °C with 5% CO2. Cells were transiently co-transfected with the human cDNA encoding the Nav1.4 α-subunit or NavMs, the β1-subunit, and eGFP. Transfection was done according to the PolyFect (Qiagen, Germantown, MD, USA) transfection protocol. A minimum of 8-hour incubation was allowed after transfection with Nav1.4 constructs. After transfection with NavMS constructs, cells were incubated between 24 – 48 hours. The cells were subsequently dissociated with 0.25% trypsin−EDTA (Life Technologies, Thermo Fisher Scientific) and plated on sterile coverslips.

Whole-cell patch clamp recordings from NavMs or Nav1.4 expressed in HEK293T cells were made using an extracellular solution composed of NaCl (140 mM), KCl (4 mM), CaCl2 (2 mM), MgCl2 (1 mM), HEPES (10 mM). The extracellular solution was titrated to pH 7.4 with NaOH. Pipettes were fabricated with a P-1000 puller using borosilicate glass (Sutter Instruments, CA, USA), dipped in dental wax to reduce capacitance, then thermally polished to a resistance of 1.5−2.5 MΩ. Pipettes were filled with intracellular solution, containing: CsF (120 mM), CsCl (20 mM), NaCl (10 mM), HEPES (10 mM) titrated to pH 7.4with CsOH.

All recordings were made using an EPC-9 patch-clamp amplifier (HEKA Elektronik, Lambrecht, Germany) digitised at 20 kHz via an ITC-16 interface (Instrutech, Great Neck, NY, USA). Voltage clamping and data acquisition were controlled using PatchMaster/FitMaster software (HEKA Elektronik, Lambrecht, Germany) running on an Apple iMac (Cupertino, California). Current was low-pass-filtered at 5 kHz. Leak and capacitance currents were subtracted automatically using a P/4 procedure following the test pulse. Gigaohm seals were allowed to stabilise in the on-cell configuration for 1 min prior to establishing the

whole-cell configuration. Series resistance was less than 5 MΩ for all recordings. All data were acquired at least 5 min after attaining the whole-cell configuration, and cells were allowed to incubate 5 min after drug application prior to data collection. Before each protocol, the membrane potential was hyperpolarised to -160 mV for NavMS constructs and -130 mV for Nav1.4 constructs to insure complete removal of both fast-inactivation and slow-inactivation. All experiments were conducted at 22 °C.

**Activation protocols.** To determine the voltage-dependence of activation in NavMs and Nav1.4, we measured the peak current amplitude at test pulse voltages ranging from -130 to +80 mV in increments of 10 mV for 19 ms. Channel conductance (G) was calculated from peak $I_{Na}$:

$$G_{Na} = I_{Na}/(V - E_{Na}) \qquad (2)$$

where $G_{Na}$ is conductance, $I_{Na}$ is peak sodium current in response to the command potential V, and $E_{Na}$ is the Nernst equilibrium potential. The midpoint and apparent valence of activation were derived by plotting normalised conductance as a function of test potential. Data were then fitted with a Boltzmann function:

$$G/G_{max} = 1/(1 + \exp(-ze_0(V_m - V_{1/2})/kT)) \qquad (3)$$

where $G/G_{max}$ is normalised conductance amplitude, Vm is the command potential, z is the apparent valence, $e_0$ is the elementary charge, $V_{1/2}$ is the midpoint voltage, k is the Boltzmann constant, and T is temperature in K.

**Steady-state inactivation protocols.** The voltage-dependence of inactivation in NavMs and fast-inactivation in Nav1.4 was measured by preconditioning the channels to a hyperpolarising potential (-130 mV for Nav1.4; -160 mV for NavMS)to insure complete channel availability, and then eliciting pre-pulse potentials that ranged from -170 to +10 mV in increments of 10 mV for 500 ms, followed by a 10 ms test pulse during which the voltage was stepped to 0 mV for Nav1.4, -20 mV for NavMS. Normalised current amplitude as a function of voltage was fit using the Boltzmann function:

$$I/I_{max} = 1/(1 + \exp(-ze_0(V_m - V_{1/2})/kT)) \qquad (4)$$

where $I_{max}$ is the maximum test pulse current amplitude. z is apparent valence, $e_0$ is the elementary charge, Vm is the prepulse potential, $V_{1/2}$ is the midpoint voltage of SSI, k is the Boltzmann constant, and T is temperature in K.

**Recovery from inactivation protocols.** The rate of recovery from inactivation was determined for NavMS constructs by holding the membrane potential at -130 mV, followed by a 500 ms depolarising pulse to 0 mV to inactivate the channels. The channels were allowed to recover from inactivation for durations between 0 and 5 s. Recovery (i.e., available channels) was then measured from a depolarising 10 ms test pulse to 0 mV. Time constants of inactivation recovery were fit to a double exponential equation:

$$I = Iss + \alpha1 \exp(-t/\tau1) + \alpha2 \exp(-t/\tau2) \qquad (5)$$

where I is current amplitude, Iss is the plateau amplitude, α1 and α2 are the amplitudes at time 0 for time constants τ1 and τ2, and t is time.

**Persistent current protocols.** Late sodium current in Nav1.4 was measured between 45 and 50 ms during a 50 ms or between 145 and 150 ms during a 200 ms depolarising pulse to 0 mV from a holding potential of -130 mV. Fifty pulses were averaged to increase signal to noise ratio[70,71].

**Statistical Analysis.** Statistical analyses were computed using Igor Pro software (WaveMetrics). Statistical analysis for NavMS and Nav1.5 was completed using one-way ANOVA followed by post-hoc Tukey test. Statistical analysis of NavMS T207A was completed using unpaired Student's t-tests. For all analysis p < 0.05 was considered statistically significant. Statistical analysis was completed by comparing test conditions to the negative control (i.e., the vehicle that the drug was dissolved in). Values are written as mean ± S.E.M., where n ≥ 3. N values represent biological replicates of individual HEK293T cells.

## Molecular Dynamics

**Molecular flooding simulations.** Simulation setups were built using the Membrane Builder and Multicomponent Assembler modules from CHARMM-GUI[72–76]. Riluzole was parameterised using CHARMM-GUI's ligand reader and modeller[77,78]. All systems were built into a 100% POPC membrane with TIP3P water, in the presence of 300 mM riluzole and sodium (Na⁺) and chlorine (Cl⁻) ions are added such that the final salt concentration is 150 mM. The simulations were carried out using GROMACS 2021 and the CHARMM36m force field was used throughout[79,80]. Periodic boundary conditions were used. Electrostatic interactions were calculated using Particle mesh Ewald (PME) and a cutoff of 12 Å used for van der Waals interactions was used. Hydrogen-containing bonds were restrained using LINCS[81]. The system was minimised using steepest descent, then six equilibration steps were ran. The first two of six equilibration steps are NVT (1 fs time step) and the final four are NPAT simulations (1 fs time step increasing to 2 fs for the final three equilibration steps)[74]. Production runs were performed in the NPT ensemble and the systems were run for a minimum of 2 μs with a time step of 2 fs. All steps were ran at 310 K. The Nosé-Hoover thermostat[82] was used with separate thermostat groups assigned for the solute, POPC membrane and the solvent. The Parrinello Rahman barostat[83] was used at a pressure of 1 atm.

Unbiased simulations were also run, placing one riluzole molecule at the entrance to the fenestration in the starting structure. This was achieved by replacing POPC molecules with a riluzole molecule. The simulation was then minimised, equilibrated and run using the same MD protocol as previously outlined but for a total of 771 ns for WT Nav1.4 and 569 ns for F1586A.

**Molecular dynamics analyses.** Clustering analysis of the simulations was carried out using the GROMACS 2021 cluster tool with the gromos clustering algorithm selected[84] and an rmsd of 2 Å used throughout. The remaining analyses in this manuscript were carried out using Python 3.7 and the Python package MDAnalysis[85,86].

To find the drug that had the highest binding with each domain, a contact analysis was performed. A binding event was considered only if the distance between the C2 or C5 of riluzole and threonine 176 was below 6 Å as well as the distance between the fluorine atoms and methionine 204 being below 6 Å. A cut-off distance of 6 Å was selected after carrying out a radial distribution function analysis of the first 200 ns of the unbiased WT simulation. The output rdf graph is shown in Supplementary Information Fig. 8, with the first plateau of contacts being at 6 Å.

Once the drug with the highest binding had been selected, the average number of contacts between the drug and M204 or T176 per time step was calculated. For this calculation, a contact was also defined if the distance between the drug atoms (C2 or C5 and F) and T176 and M204 was below 6 Å. Both binding site residues needed to be in contact with the drug for it to be considered in contact with the binding site. The sum of all the contacts over time for each drug atom was calculated and then divided by the total length of the trajectory

from which the analysis was run, to obtain the average number of contacts per time step. This distance cut-off was also used to calculate a residency time, counting the number of consecutive frames that both binding constraints were satisfied. The maximum residency time for each mutant is reported.

## Reporting summary
Further information on research design is available in the Nature Portfolio Reporting Summary linked to this article.

## Data availability
The atomic coordinates and crystallographic structure factors generated for NavMs complexed with riluzole have been deposited in the protein data bank (PDB) under the accession code 8S6J with data collection and refinement statistics details provided in Supplementary Information Table 1. The PDB code of the previously published structure used in this study is 6SX5. Source data for NMR and electrophysiology performed in this study are provided with this paper as a Source Data file. The MD simulation data sets with initial and final system structures of all six molecular dynamics experiments along with forcefield parameters are deposited in the Zenodo database with the identifier [https://doi.org/10.5281/zenodo.13293736] Source data are provided with this paper.

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

## Acknowledgements

We thank the staff of Diamond Light Source (data obtained under proposal MX 23853), the European Synchrotron Radiation Facility (Grenoble, data obtained under proposal MX 2415) and EMBL (DESY, Hamburg, data obtained under proposal MX 842) for assistance and support on the beamlines used in this study. This study was supported by grants from the Rosetress Trust and the UK Biological and Biotechnology Research Council (BBSRC) to BAW; BBSRC Standard Grant BB/S017844/1 to MBU; a joint travel grant to BAW and PCR from the UK BBSRC; Beamtime grants at Diamond Light Source to BAW, AS and DH; This work was also supported by the Francis Crick Institute through the provision of access for DH and BAW to the MRC Biomedical NMR Centre, which is funded by Cancer Research UK (CC1078), the UK Medical Research Council (CC1078), and the Wellcome Trust (CC1078); a Wellcome Trust Studentship to FT; a BBSRC London Interdisciplinary PhD Programme (LIDo) studentship to RL-RDC; Discovery Grants from the Natural Science and Engineering Council of Canada to DAP and MAF.

## Author contributions

D.H., F.T. and D.A.P. contributed equally to this work. B.A.W. and D.H. conceived and planned the project. D.H. performed the molecular biology, protein expression and purification for all structural studies, D.H., A.S. and V.B.M. performed and analysed crystallography, G.K. and D.H. performed and analysed N.M.R., F.T. performed all M.D. simulations, F.T. and R.L.-R.D.C. performed the simulation analysis under the supervision of M.B.U., D.A.P. (NavMs) and M.A.F. (Nav1.4) performed and analysed electrophysiology under the supervision of P.C.R. D.H. and F.T. wrote the manuscript under the supervision of B.A.W. and with input from all authors All authors approved of the final version.

## Competing interests

The authors declare no competing interests.
