## [Peer Review File · Nature Communications]

Structural basis for the rescue of hyperexcitable cells by the Amyotrophic Lateral Sclerosis drug RiluzoleREVIEWER COMMENTS

Reviewer #1 (Remarks to the Author):

Hollingworth et al. studied the structural basis of riluzole interaction with NavMs as a model of neuronal sodium channels. They found an atypical binding site of riluzole which may explain how this neuroprotective ALS drug stabilizes VGSCs in their inactivated state, which presumably blocks late sodium current. They also performed MD simulations to study how Riluzole accesses VGSCs and does its therapeutic action. They discovered that it binds in the fenestrations of NavMs and stabilizes the inactivated state of the channel plausibly by affecting the amino acids network that talks to the inactivation gate. They also studied riluzole effects on a disease variant of the VGSC isoform Nav1.4, where pathologically increased late sodium current is induced by a disease mutation. The structural and functional work has been well executed. The manuscript is well written, however, there are still concerns about the rationale of this manuscript.

Major concerns:

1- Riluzole blocks late sodium current specifically without affecting peak sodium current, however, the authors used the WT NavMs, which has no intrinsic late sodium current, as a model for studying the drug's effect. In addition, bacterial sodium channels in general lack the fast inactivation machinery that exists in their eukaryotic counterparts. The question is how the authors could map the structural basis of riluzole interaction with NavMs to the proposed action of the drug on eNavs.

2- Another drug that is well-known for blocking late sodium current is ranolazine. A recent paper showed how ranolazine interacts with cardiac sodium channel Nav1.5 (Lenaeus et al. 2023). The authors should discuss the differences/similarities between how ranolazine interacts with Nav1.5 and how riluzole binds to NavMs. There should be common features of this interaction. An overlay of the two structures would help to elucidate these features which would help in designing a new generation of anti-late sodium current drugs. This comparison may also justify their use of NavMs as a model for eukaryotic sodium channels.

Reviewer #2 (Remarks to the Author):

This paper by Hollingworth and colleagues presents structural studies of the binding of the drug riluzole to a bacterial homolog of voltage-gated sodium channels (VGSCs).

Riluzole is a hydrophobic compound used in the therapy of amyotrophic lateral sclerosis (ALS), a neurodegenerative disease for which there is no cure. Riluzole significantly increases life expectancy of ALS patients at a micromolar dose and binds to several targets including VGSCs of which it stabilizes a non-conducting inactivated state. Structural basis for this inhibition is still missing.

The authors here use a bacterial homolog of eukaryotic VGSC, a BacNav called NavMs, that is more suitable for structural characterization.

They first verify that riluzole binds to NavMs using STD NMR before describing X-ray crystallography

work of NavMs in complex with riluzole. The crystallography data is very compelling and include control datasets using riluzole-solvent DMSO and SAD data collection to detect the sulphur atom of the compound. They unambiguously place riluzole in a hydrophobic pocket close to the ion channel, referred as lateral fenestration. They verify this binding mode in solution by performing STD NMR on NavMs labelled by BTFA in the pocket (M204).

They then investigate the functional effect of riluzole on NavMs by whole-cell electrophysiology in transfected HEK cells. They could recapitulate the mode of action of the drug on the bacterial channel (hyperpolarization of the SSI and slowed RFI without change in the activation kinetics and no channel block up to 25 μ M). This characterization is consistent with the binding site they identify in the X-ray structure and further demonstrate the relevance of NavMs as a model for eukaryotic VGSCs.

The authors further investigate riluzole binding in molecular dynamics, which shows a binding consistent with X-ray data in the fenestrations close to the pore.

Residues involved in binding of riluzole are homologous to those forming the local anesthetic binding site in the eukaryotic VGSC hNav1.4, notably lined by residue Phe1586. To further demonstrate that this pocket is the one targeted by riluzole in human VGSCs, they recorded electrophysiological profile of riluzole on the WT and F1586A mutant hNav1.4. The mutation abolishes riluzole hyperpolarization of the SSI. In parallel, they investigate the effect of the mutation of Thr207 of NavM, a residue lining the binding pocket and homologous to Phe1586. They found that NavMs T207A is also not sensitive to riluzole.

To go further, they performed molecular dynamics on a "humanized" T207F NavMs and they found a higher residence time for riluzole in the binding site as compared to NavM WT, consistent with the known affinities for riluzole of both proteins.

Finally, the authors performed molecular dynamics simulation of riluzole binding to the human Nav1.4 (using its previously published cryo-EM structure). The first experiment shows a binding further away from the pore and away from Phe1586. The authors interpret this result in regards of the presence of a lipid in the fenestration pocket. They ran again the simulation but this time placing the riluzole closer to the pocket entry ; a run for which they see a binding mode in the fenestration compatible with the rest of their data.

They finish the study by examining the effect of riluzole on a hNavA.4 disease variant found in patients with myotonia.

Riluzole is an important drug used in a prevalent disease and its mechanisms of action are important to uncover. Even if the authors use a bacterial model of the human channel, they make a strong demonstration that this model is relevant.

The experiments are sound and compelling, the methods are detailed and clear.

I do feel that there are some questions that should be addressed/clarified before recommending publication.

1) A major point is the absence of discussion about the conformation of the channel, both in the structure and the MD simulations.

The authors states in the introduction that riluzole stabilizes an inactive conformation of the channel but their structure shows a wide-open pore. The non-expert reader must have a detailed look at the methods section to eventually learn that this conformation is considered as non-conducting. I believe this topic should be raised way before in the result section.

Overall, the study lacks a possible molecular mechanism of action of the drug. Do the authors have hypothesis on how riluzole stabilizes the inactive channel?

Giving what we know of the conformational changes occurring during activation/inactivation, what happens to the fenestration pocket in a conducting state? Is it still compatible with riluzole binding? These points must be discussed to have a comprehensive view of riluzole action on VGSCs.

2) An interesting point the authors discussed is the possible ability of riluzole to act as a pore blocker at high concentration. This pore block is seen at 100 μ M in electrophysiology and was observed for other drugs on VGSCs. I couldn't find any concentration values used for STD, did the authors performed the STD with a large excess in riluzole? In the crystals, the drug is added to \sim 1.5 fold excess. Did the authors tried to add more riluzole during crystallogenesis? This could reveal an additional pore block site. The authors should also compare their conditions to published structures with drugs bound to the pore.

3) The last section describing riluzole effect on a pathological variant is interesting but is not related to the rest of the paper. The authors should discuss the position of P1158 in regards of their structural data.

Minor comments:

1) Several figures are difficult to read, the text is too small. In the main figures: Fig 1c, 2a (protocol), 2b (axis), 2c, 2e (Hole2 analysis), 3c, 4a, 4b, 6. Figure 6b inserts are unreadable.

2) Fig 1c and Ext Fig2 lacks labels, especially of side chains of the displayed residues.

3) Fig1 should include a global unclipped view of the protein with domains labelled for the non-experts.

4) For electrophysiology data, number of cells should be indicated.

Reviewer #3 (Remarks to the Author):

This is an interesting study on the interaction of Riluzole with inactivated conformations of Nav channels. All of the structural biology utilized X-ray crystallography and the prokaryotic NavM homotetrameric channel as a model for eukaryotic pseudotetrameric channels, although both Molecular Dynamics (MD) simulations and functional electrophysiological studies were undertaken to connect work on NavM with eukaryotic Nav channels. The overall insight is that Riluzole stabilizes the inactivated state of NavM and by extension eukaryotic Nav channels by binding within a fenestration between the membrane and the aqueous ion permeation pathway. Although the path to these core conclusions is not completely straightforward, requiring the use of saturation-transfer NMR experiments and anomalous diffraction experiments, the core conclusions are supported by the data provided and are reasonable. The authors also appropriately highlight the value of Riluzole throughout the manuscript as it is a safe and approved drug by the FDA for various diseases involving hyperexcitability and incomplete Nav channel inactivation. I particularly like the data in Fig. 6 on the mutant causing myotonia. There are a few places where additional data would support the conclusions, but overall this comes across as a thoughtful study.

1) I appreciate the authors attempts to connect their work on NavM with eukaryotic Nav channels, but I found it rather curious that functional data are not provided for the T207F mutant that was explored with MD. Knowing what this mutation does to the gating properties and the ability of

Riluzole to stabilize an inactivated state of NavM would be an extremely worthwhile addition to the manuscript.

2) I also found it curious that the authors don't say anything about how the Riluzole binding site within the fenestrations changes conformation as Nav channels gate to better understand why this drug stabilizes the inactivated state. It would certainly be interesting to compare the Riluzole binding site for all available conformations of NavM, other prokaryotic Nav channels and eukaryotic Nav channels.

3) Recent work on inactivation of Kv2.1 channels has proposed that the channel inactivates by closure of the internal pore (PMC10567553), being regulated by a hydrophobic coupling nexus not far from where Riluzole binds in NavM channels. Moreover, a recent study on inactivation in eukaryotic Nav channels identified residues that line the internal pore in structures thought to be inactivated where truncating double mutants produced a leaky inactivated state (PMC10442390). The residues lining the inactivated internal pore in these two studies are at similar positions within the S6 helices, suggesting that inactivation by closure of the internal pore might be similar between Nav channels and Kv channels. The authors might consider bringing in these studies in their discussion.

4) Some of the figures could be improved by making text and data more uniform in size. For example, the STD data in Fig.1C needs to be properly presented in a much larger layout and most of the text in graphs throughout the figures needs to be substantially larger to be legible without magnification.

Reviewer #4 (Remarks to the Author):

As requested, I am writing this review on my expertise in molecular dynamics of ion channels.

Therefore, I only judge the MD part of the manuscript.

Overall, the authors performed 2 μ s unbiased MD flooding simulations with 300 μ M riluzole using Charmm-gui to setup the simulations and the CHARMM36m force field to run the simulations. All MD parameters and setups are valid and meet the expected standards in the field. Simulation setups and analysis scripts are made publicly available, enabling straightforward reproducibility of the simulations. Importantly, MD simulations could reproduce the drug orientation observed experimentally, further strengthening the validity of the simulations.

What is not really clear to me is the exact mechanism, how drug binding to the fenestrations allosterically stabilizes channel inactivation? But this perhaps will require much longer simulations, done in a follow-up publication.

To the Reviewers:

Below are our point-by-point responses to the reviewers' questions for our paper "Structural basis for rescue of hyperexcitable cells by the ALS drug riluzole" (manuscript 14662 by Hollingsworth et al) [text in brown = comments from the reviewers to the authors; text in green = replies from the authors; text in red= modified text in in the manuscript in response to reviewers' comments].

Reviewer #1 Comments (Remarks to the Authors):

Hollingsworth et al. studied the structural basis of riluzole interaction with NavMs as a model of neuronal sodium channels. They found an atypical binding site of riluzole which may explain how this neuroprotective ALS drug stabilizes VGSCs in their inactivated state, which presumably blocks late sodium current. They also performed MD simulations to study how Riluzole accesses VGSCs and does its therapeutic action. They discovered that it binds in the fenestrations of NavMs and stabilizes the inactivated state of the channel plausibly by affecting the amino acids network that talks to the inactivation gate. They also studied riluzole effects on a disease variant of the VGSC isoform Nav1.4, where pathologically increased late sodium current is induced by a disease mutation. The structural and functional work has been well executed. The manuscript is well written, however, there are still concerns about the rationale of this manuscript.

General Response to Reviewer 1: We very much appreciate the reviewer's evaluation and positive comments regarding this study.

Reviewer #1 Response Comment to Authors Comment 1 and subsequent revision:

Reviewer Comment 1- Riluzole blocks late sodium current specifically without affecting peak sodium current, however, the authors used the WT NavMs, which has no intrinsic late sodium current, as a model for studying the drug's effect. In addition, bacterial sodium channels in general lack the fast inactivation machinery that exists in their eukaryotic counterparts. The question is how the authors could map the structural basis of riluzole interaction with NavMs to the proposed action of the drug on eNavs.

Authors' Reply (part a): We agree that this is a very important point. NavMs, like all BacNavs, lacks the inter-domain loop element that is prerequisite for the fast inactivation of eNavs, showing only slow inactivation. However, riluzole has been shown to affect the fast and slow inactivation of eNavs with equivalent potencies (IC50s for both mirroring the potency seen at therapeutic concentrations for riluzole's effect on slow inactivation with NavMs), indicative of the involvement of a common binding event. We had removed 2 lines and the reference from the evolving manuscript which covered this point (below) which we have now reinstated.

Revised text: "As a model for eukaryotic inactivation BacNavs only have slow inactivation processes reminiscent of slow inactivation in eNavs. However, it has been shown that riluzole modulates both fast and slow inactivation with equal potency indicative of a common binding event inhibiting both processes." (PMID: 9262334)

Author's Reply (part b): In addition, as noted in the text, all the electrophysiological parameters that we could measure with riluzole that are common to BacNavs and eNavs proved remarkably similar, as were the binding sites found for the drug in Nav1.4 complimentary MD work with Nav1.4, which provides our justification for using this system. No change is needed to text as this is noted already in the manuscript.

Reviewer #1 (Response to Comment 2): -

Another drug that is well-known for blocking late sodium current is ranolazine. A recent paper showed how ranolazine interacts with cardiac sodium channel Nav1.5 (Lenaeus et al. 2023). The authors should discuss the differences/similarities between how ranolazine interacts with Nav1.5 and how riluzole binds to NavMs. There should be common features of this interaction. An overlay of the two structures would help to elucidate these features which would help in designing a new generation of anti-late sodium current drugs. This comparison may also justify their use of NavMs as a model for eukaryotic sodium channels.

Reply: We thank the reviewer for this suggestion. We note that the binding sites of many voltage-gated ion channel drugs are rapidly emerging, mostly facilitated by cryoEM. The ranolazine paper mentioned above is one such paper. However, one interesting factor in our paper, is that we uncovered a function-affecting binding site for a drug, (a small drug of only 20 atoms and few pharmacophores, but which shows high potency towards VGSCs); that site is unlikely to be revealed by cryoEM due the interaction of competing lipids at the same site and maybe also because of the deterioration of data quality on prolonged exposure to the drug as reported in the Yan paper we reference (PMID: 37782796). In our case, although we achieved only ~20% occupancy for riluzole at the site, we also used long-wavelength crystallography and site-specific NMR to unambiguously identify and place the drug. Although Ranolazine effects on VGSCs share similarity with riluzole it is a much bigger drug (64 atoms and many more pharmacophores); and in this paper it is found binding in the pore, where mutation of residues within the identified site is shown to compromise tonic block and use-dependent block, two effects on VGSCs which are lacking at the therapeutic riluzole concentrations we have explored. We think that this ranolazine study, alongside the other recent structural studies of VGSCs-targeting drugs, and our work, will allow for an interesting near-future review to discuss this issue, but we believe that introducing it here would not promote understanding of the model we have based on our clear experimental data. No change made to the text.

Reviewer #2: (General Responses to the Reviewer's comments)

This paper by Hollingworth and colleagues presents structural studies of the binding of the drug riluzole to a bacterial homolog of voltage-gated sodium channels (VGSCs).

Riluzole is a hydrophobic compound used in the therapy of amyotrophic lateral sclerosis (ALS), a neurodegenerative disease for which there is no cure. Riluzole significantly increases life expectancy of ALS patients at a micromolar dose and binds to several targets including VGSCs of which it stabilizes a non-conducting inactivated state. Structural basis for this inhibition is still missing.

The authors here use a bacterial homolog of eukaryotic VGSC, a BacNav called NavMs, that is more suitable for structural characterization.

They first verify that riluzole binds to NavMs using STD NMR before describing X-ray crystallography work of NavMs in complex with riluzole. The crystallography data is very compelling and include control datasets using riluzole-solvent DMSO and SAD data collection to detect the sulphur atom of the compound. They unambiguously place riluzole in a hydrophobic pocket close to the ion channel, referred as lateral fenestration. They verify this binding mode in solution by performing STD NMR on NavMs labelled by BTFA in the pocket (M204).

They then investigate the functional effect of riluzole on NavMs by whole-cell electrophysiology in transfected HEK cells. They could recapitulate the mode of action of the drug on the bacterial channel (hyperpolarization of the SSI and slowed RFI without change in the activation kinetics and no channel block up to 25µM). This characterization is consistent with the binding site they identify in the X-ray structure and further demonstrate the relevance of NavMs as a model for eukaryotic VGSCs.

The authors further investigate riluzole binding in molecular dynamics, which shows a binding consistent with X-ray data in the fenestrations close to the pore.

Residues involved in binding of riluzole are homologous to those forming the local anesthetic binding site in the eukaryotic VGSC hNav1.4, notably lined by residue Phe1586. To further demonstrate that this pocket is the one targeted by riluzole in human VGSCs, they recorded electrophysiological profile of riluzole on the WT and F1586A mutant hNav1.4. The mutation abolishes riluzole hyperpolarization of the SSI. In parallel, they investigate the effect of the mutation of Thr207 of NavM, a residue lining the binding pocket and homologous to Phe1586. They found that NavMs T207A is also not sensitive to riluzole.

To go further, they performed molecular dynamics on a “humanized” T207F NavMs and they found a higher residence time for riluzole in the binding site as compared to NavM WT, consistent with the known affinities for riluzole of both proteins.

Finally, the authors performed molecular dynamics simulation of riluzole binding to the human Nav1.4 (using its previously published cryo-EM structure). The first experiment shows a binding further away from the pore and away from Phe1586. The authors interpret this result in regards of the presence of a lipid in the fenestration pocket. They ran again the simulation but this time placing the riluzole closer to the pocket entry ; a run for which they see a binding mode in the fenestration compatible with the rest of their data.

They finish the study by examining the effect of riluzole on a hNavA.4 disease variant found in patients with myotonia. **Riluzole is an important drug used in a prevalent disease and its mechanisms of action are important to uncover. Even if the authors use a bacterial model of the human channel, they make a strong demonstration that this model is relevant.**

The experiments are sound and compelling, the methods are detailed and clear.

Reply: We greatly appreciate the reviewer’s highly supportive evaluation of our work and these positive comments. No changes required to the manuscript text.

Reviewer# 2 (specific comments): I do feel that they are some questions that should be addressed/clarified before recommending publication:

1) A major point is the absence of discussion about the conformation of the channel, both in the structure and the MD simulations.

The authors states in the introduction that riluzole stabilizes an inactive conformation of the channel but their structure shows a wide-open pore. The non-expert reader must have a detailed look at the methods section to eventually learn that this conformation is considered as non-conducting. I believe this topic should be raised way before in the result section.

Replies: We agree with the reviewer and thank them for pointing this out. The explanatory sentence that is referred to which was in the methods section, below, has been moved to the main section of the paper:

“The apo-structure was initially reported to be in the open state due to its wide pore gate. However, recent work has shown this channel to be non-conductive and also convincingly proposes that π -helix driven transitions in the channel pore, (not present in the NavMs structure), are required for Na^+ conduction redefining this structure as representing an inactivated channel state” (PMID: 34890580).

For Nav1.4, in order to make this clearer we have also changed the sentence “We used the same MD simulation as before but replaced NavMs with hNav1.4 (the first available cryoEM structure for an hNav, PDB ID: 6AGF⁵⁴, inactivated state)” to :

“We used the same MD simulation as before but replaced NavMs with hNav1.4 (the first available cryoEM structure for an hNav, 9PDB ID: 6AG)F⁵⁴, also captured in the inactivated state)

Reviewer General Comments: Overall, the study lacks a possible molecular mechanism of action of the drug. Do the authors have hypothesis on how riluzole stabilizes the inactive channel? Given what we know of the conformational changes occurring during activation/inactivation, what happens to the fenestration pocket in a conducting state? Is it still compatible with riluzole binding? These points must be discussed to have a comprehensive view of riluzole action on VGSCs.

Reply: We thank the reviewer for these two comments and will answer them together as our response below is related to both points:

The complication of trying to understand how voltage-gated ion channels work in a polarised membrane when all near-atomic resolution structural work is only possible at 0mV, means that many theories/models exist as to what 'gating' looks like. The S6 π -helix model for NavMs conduction mentioned in the response to point 2 above, allows conduction of sodium ions through NavMs, where the structural rearrangements involved in the transition also narrow/close the fenestrations. Fenestration closure is also found in a complimentary simulation that allows the BacNav, NavAb to conduct Na⁺, (PMID: 36515966). π -helices have also been found in some S6 helices of eNav cryoEM structures. In the first cryoEM structure of an eNav (Cockroach, PDB IB: 5X0M) all 4 S6 helices contain π helical regions and all fenestrations are severely narrowed or closed. A follow-up publication by the same group uses MD simulations to propose how π -helical transitions could lead to Na⁺ conduction in eNavs. (PMID: 37341700)

Closing the fenestration in the process of channel opening will empty it of lipid, and it is intriguing that in our MD simulation with Nav1.4 and the DIII-DIV fenestration, where we begin with an empty fenestration and with a riluzole molecule in the membrane near the fenestration opening (in a position allowing it to compete with lipid for fenestration binding, rather than it starting in the aqueous phase), riluzole binds rapidly and sustainably. Riluzole binding could stabilise the inactivated fenestration-accessible channel state, by this binding event inhibiting this transition. However, we believe that this mechanism (as well as others we could have made a case for such as the potential effects of riluzole binding on disruption of the lipid network) all make a lot of assumptions about gating that need more experimental justification. Thus, we concentrated on what we found experimentally and avoided potential controversy and error associated with pushing one molecular mechanism of action of the drug over another, even though we were tempted to do so. Having said that, it is necessary for us to present this as potential gating-mechanism we would add the model above into the discussion as we personally believe it has merit, although it currently lacks certainty. We feel all of the other data is direct and have tried to avoid such speculation that is unsupported by data.

Reviewer Comment: An interesting point the authors discussed is the possible ability of riluzole to act as a pore blocker at high concentration. This pore block is seen at 100 μ M in electrophysiology and was observed for other drugs on VGSCs. I couldn't find any concentration values used for STD, did the authors performed the STD with a large excess in riluzole? In the crystals, the drug is added to ~1.5 fold excess. Did the authors tried to add more riluzole during crystallogenesis? This could reveal an additional pore block site. The authors should also compare their conditions to published structures with drugs bound to the pore.

Reply: We thank the reviewer for pointing out this missing information. It was an oversight by us. The saturating concentration of riluzole in our buffer was 0.5mM and we used this for both STD NMR and crystallography. The protein concentration in the STD experiments was 50 μ M (monomer concentration). **We have added this information regarding concentrations to the STD NMR section of the methods.**

It is true that for the early NavMs 19F_Phe experiment both a pore site and the fenestration site could be represented by the overall STD. But this was the earliest experiment where we were looking for any interaction in order to have a reason to proceed with crystallography. Of note, a

pore binding-site would not be reported by the specific NavMs M204C_BTFA STD NMR experiment, as it would be too far away.

Reviewer Comment: *The last section describing riluzole effect on a pathological variant is interesting but is not related to the rest of the paper. The authors should discuss the position of P1158 in regards of their structural data.*

Reply: We did think of doing this, but authors agreed that the positioning of the mutation is not important to this investigation and so highlighting it would potentially mislead readers into thinking that the site of mutation, rather than the consequence, was important. The mutations/cell damage that leads to ALS causes VGSCs to produce pathological INaL indirectly, while this is a mutation, within the same channel we used in our MD experiments as the model for eNavs, where a direct mutation within Nav1.4 also causes pathological INaL which can be reversed by riluzole. Further justification for not flagging-up the position of the mutation was that riluzole has been shown to be effective against 'epilepsy' mutations in Nav1.6 that enhance persistent sodium current and where mutations occur at different regions of the protein. (PMID: 32968789).

Reviewer Minor Comments:

1) Several figures are difficult to read, the text is too small. In the main figures: Fig 1c, 2a (protocol), 2b (axis), 2c, 2e (Hole2 analysis), 3c, 4a, 4b, 6. Figure 6b inserts are unreadable.

2) Fig 1c and Ext Fig2 lacks labels, especially of side chains of the displayed residues.

3) Fig1 should include a global unclipped view of the protein with domains labelled for the non-experts.

4) For electrophysiology data, number of cells should be indicated.

Reply: We agree and have addressed these minor comments in the revised article with either figure revisions or more descriptive figure legends. Number of cells in the electrophysiology data are indicated in many cases, but in some cases where we show data from different concentrations etc, they can be found in the supplementary table, as directed to in the text.

When producing a figure revision to minor comment 3, this led to an awkward figure. But we do acknowledge the importance of this general point and so have now incorporated an extra figure into the paper (now figure 1, shown below for convenience), to familiarise non-experts with VGSC structure and the differences between structures of BacNav's and eNav's. We thank the reviewer for making us aware of this, as we think this improves the paper.

Figure 1: Prokaryotic VGSCs (BacNavs) are structurally simpler compared to Eukaryotic VGSCs (eNavs) but share basic functional architecture. a) Basic topologies of the single chains of bacterial NavMs and eukaryotic Nav1.4, b) Four individual subunits of NavMs produce the functional homotetrameric channel, c) Nav1.4 folds from the single chain to form the functional pseudo-tetrameric channel.

Reviewer #3 (Remarks to the Author):

This is an interesting study on the interaction of Riluzole with inactivated conformations of Nav channels. All of the structural biology utilized X-ray crystallography and the prokaryotic NavM homotetrameric channel as a model for eukaryotic pseudotetrameric channels, although both Molecular Dynamics (MD) simulations and functional electrophysiological studies were undertaken to connect work on NavM with eukaryotic Nav channels. The overall insight is that Riluzole stabilizes the inactivated state of NavM and by extension eukaryotic Nav channels by binding within a fenestration between the membrane and the aqueous ion permeation pathway. Although the path to these core conclusions is not completely straightforward, requiring the use of saturation-transfer NMR experiments and anomalous diffraction experiments, the core conclusions are supported by the data provided and are reasonable. The authors also appropriately highlight the value of Riluzole throughout the manuscript as it is a safe and approved drug by the FDA for various diseases involving hyperexcitability and incomplete Nav channel inactivation. I particularly like the data in Fig. 6 on the mutant causing myotonia. There are a few places where additional data would support the conclusions, but overall this comes across as a thoughtful study.

Reply: We thank the reviewer for both these initial overall comments and the positive evaluation of our work.

1) I appreciate the authors attempts to connect their work on NavM with eukaryotic Nav channels, but I found it rather curious that functional data are not provided for the T207F mutant that was explored with MD. Knowing what this mutation does to the gating properties and the ability of Riluzole to stabilize an inactivated state of NavM would be an extremely worthwhile addition to the manuscript.

Reply: We agree with this and would have liked to do this. However, due to the homotetrameric nature of BacNavs, introducing phenylalanine at this site on the S6 helix actually introduces 4 Phe residues into the pore, instead of the single Phe found in eNavs (the other eNav domains containing serine (x2) or asparagine at the equivalent positions). Previous attempts at electrophysiology using BacNavs and the equivalent mutant in NavAb (T206F) and NachBac (T220F) yielded cells that did not conduct ionic currents. Therefore, we were limited to using an MD simulation which allowed us to place the T207F mutation into only one site in the tetramer and compare with the other sites for binding, but without the corresponding functional data.

2) I also found it curious that the authors don't say anything about how the Riluzole binding site within the fenestrations changes conformation as Nav channels gate to better understand why this drug stabilizes the inactivated state. It would certainly be interesting to compare the Riluzole binding site for all available conformations of NavM, other prokaryotic Nav channels and eukaryotic Nav channels.

Reply: We address this point in the response below.

3) Recent work on inactivation of Kv2.1 channels has proposed that the channel inactivates by closure of the internal pore (PMC10567553), being regulated by a hydrophobic coupling nexus not far from where Riluzole binds in NavM channels. Moreover, a recent study on inactivation in eukaryotic Nav channels identified residues that line the internal pore in structures thought to be inactivated where truncating double mutants produced a leaky inactivated state (PMC10442390). The residues lining the inactivated internal pore in these two studies are at similar positions within the S6 helices, suggesting that inactivation by closure of the internal pore might be similar between Nav channels and Kv channels. The authors might consider bringing in these studies in their discussion.

Reply: We thank the reviewer for this comment and pointing these papers out. However, there are quite a few separate theories and models for how gating happens, and we wanted to avoid 'picking favourites'. For our system, the transitional gating model of Choudhury et al (PMID: 34890580) seems very convincing especially as it involves our channel. In this model, NavMs transition from the non-conducting inactivated state found in our structure to a conducting state, through π -helix formation on the S6 helices. π -helices have been found in the S6 helices of many eNav cryoEM structures and α -to- π secondary structure transitions in S6 are widely accepted as controlling such gating in transient receptor potential (TRP) channels. However, we did not want to "cherry pick" amongst models as they currently lack certainty in VGSCs and chose to focus on what our work showed rather than speculate. However, if required we would add our thoughts on this in the discussion.

Reply to Comment 2: The π -helix model presented by Choudhury for both NavMs and a follow up for the BacNav NavAb, for which there is also high-resolution structural information (PMID: 36515966), finds that the structural transitions involved in moving from the non-conducting to conducting state, results in closure of the fenestrations in the conducting open-channel. These fenestrations would be empty of both lipid and drug and it is interesting that when we ran the MD simulation to investigate a Nav1.4 DIII-DIV fenestration site starting a riluzole molecule close to an lipid-empty fenestration, riluzole bound rapidly and sustainably. π -helices have been found in the S6 helices of many cryoEM eNav sequences. Including in all 4 S6 helices for the cockroach sodium channel which was the first to be solved, and where all fenestrations are closed. The group responsible for this work on BacNavs have also presented MD simulations for eNav channels showing how similar transitioning can lead to Na^+ conduction in these channels, (PMID: 37341700).

However, many other models exist on gating and we wanted to avoid any controversy associated with selecting a single model in such an uncertain environment.

Comment 4) Some of the figures could be improved by making text and data more uniform in size. For example, the STD data in Fig.1C needs to be properly presented in a much larger layout and most of the text in graphs throughout the figures needs to be substantially larger to be legible without magnification.

Reply: We agree with the reviewer and have improved the figures accordingly. For convenience, the updated Figure 1 (now Figure 2) is presented below:

Reviewer #4 (Remarks to the Authors):

Comments: As requested, I am writing this review on my expertise in molecular dynamics of ion channels. Therefore, I only judge the MD part of the manuscript. Overall, the authors performed 2μs unbiased MD flooding simulations with 300μM riluzole using Charmm-gui to setup the simulations and the CHARMM36m force field to run the simulations. All MD parameters and setups are valid and meet the expected standards in the field. Simulation setups and analysis scripts are made publicly available, enabling straightforward reproducibility of the simulations. Importantly, MD simulations could reproduce the drug orientation observed experimentally, further strengthening the validity of the simulations.

Reply: We very much appreciate these reviewer comments regarding our MD simulation work

What is not really clear to me is the exact mechanism, how drug binding to the fenestrations allosterically stabilizes channel inactivation? But this perhaps will require much longer simulations, done in a follow-up publication.

Reply: Because voltage across a membrane is required for a VGSC to transition between states the structural basis for VGSC gating is not clear. However, MD simulations have been attempted, and in one such study using NavMs, a S6 π-helix transition is proposed to facilitate Na⁺ conduction (Choudhury et al, PMID: 34890580). This transition of the S6 helix closes the fenestrations in the ‘open state’. Riluzole binding within the fenestration in the inactivated state, may provide a

simple mechanism for stabilisation of this state. Using similar techniques, the same group have shown how π -helix transitions can facilitate conduction in the BacNav, NavAb, (PMID: 36515966) and eNavs (PMID: 37341700). However, although we could use this as the basis for a mechanism of action, many other theories/models exist for VGSC gating, and none are structurally proved, so we avoided speculation and any controversy associated with 'cherry-picking' one gating model over another, preferring to concentrate on the results we attained in what we consider a rounded study.

REVIEWERS' COMMENTS

Reviewer #1 (Remarks to the Author):

The authors addressed my concerns in the revisions.

Reviewer #2 (Remarks to the Author):

The authors have answered all my questions and remarks, I recommend publication of the manuscript

Reviewer #3 (Remarks to the Author):

The authors have revised the manuscript to address a few of the comments of the reviewers, but for the most part have explained/argued why they prefer to maintain a prokaryotic centric perspective on Nav channels inactivation mechanisms. It's a shame because embracing the comments of the reviewers would have made the work more relevant for future studies on eukaryotic channels.

Reviewer #4 (Remarks to the Author):

I have no criticism on the revised version of the manuscript.